# A first order phase transition mechanism underlies protein aggregation in mammalian cells

Arjun Narayanan[1], Anatoli Meriin[2], J Owen Andrews[1], Jan-Hendrik Spille[1], Michael Y Sherman[3], Ibrahim I Cisse[1]*

[1]Department of Physics, Massachusetts Institute of Technology, Cambridge, United States; [2]Department of Biochemistry, Boston University School of Medicine, Boston, United States; [3]Department of Molecular Biology, Ariel University, Ariel, Israel

**Abstract** The formation of misfolded protein aggregates is a hallmark of neurodegenerative diseases. The aggregate formation process exhibits an initial lag phase when precursor clusters spontaneously assemble. However, most experimental assays are blind to this lag phase. We develop a quantitative assay based on super-resolution imaging in fixed cells and light sheet imaging of living cells to study the early steps of aggregation in mammalian cells. We find that even under normal growth conditions mammalian cells have precursor clusters. The cluster size distribution is precisely that expected for a so-called super-saturated system in first order phase transition. This means there exists a nucleation barrier, and a critical size above which clusters grow and mature. Homeostasis is maintained through a Szilard model entailing the preferential clearance of super-critical clusters. We uncover a role for a putative chaperone (RuvBL) in this disassembly of large clusters. The results indicate early aggregates behave like condensates.

**Editorial note:** This article has been through an editorial process in which the authors decide how to respond to the issues raised during peer review. The Reviewing Editor's assessment is that all the issues have been addressed (see decision letter).

DOI: https://doi.org/10.7554/eLife.39695.001

*For correspondence:
icisse@mit.edu

Competing interests: The authors declare that no competing interests exist.

## Introduction

Neurodegenerative diseases, such as Parkinson's disease, Amyotrophic Lateral Sclerosis, and Alzheimer's disease, are characterized by the appearance of large protein aggregates in cells and in the extracellular space (*Selkoe, 2004*). But the presence of aggregates does not always directly correlate with disease progression (*Ross and Poirier, 2004*). Intermediate species of aggregates, created early in the aggregation process, could be more toxic to cells than large aggregates, plaques and fibres (*Cookson, 2005*; *Gosavi et al., 2002*; *Karpinar et al., 2009*; *Lashuel et al., 2002a*; *Lashuel et al., 2002b*; *Pountney et al., 2004*; *Ross and Poirier, 2004*; *Xu et al., 2002*). However, the early steps of aggregate formation have been difficult to study, and may be critical to untangling the relationship between aggregation and disease

Experimentally, late stages of aggregation can be measured both in vitro and in living cells, but the very early steps of aggregate formation remain elusive due to methodological limitations. In previous studies, the dynamics of nucleation and growth (*Fink, 1998*; *Morris et al., 2009*) of protein aggregates were most commonly measured experimentally in vitro (*Buell et al., 2014*; *Jarrett and Lansbury, 1993*; *Krishnan and Lindquist, 2005*; *Lomakin et al., 1996*; *Serio et al., 2000*). Nucleation in vitro typically requires a high concentration of purified monomers, which may not represent physiological conditions in living cells. Alternatively, aggregate growth dynamics have been studied in living cells, typically by seeding preformed 'nuclei' inside the living cells (*Kaminski and Kaminski*

*Schierle, 2016*; *Nonaka et al., 2010*). These live cell experiments bypass the nucleation step altogether by externally seeding the aggregation. Thus, neither of these approaches has captured the early stages of protein aggregation that are likely to occur in unperturbed cells.

A direct experimental readout of the very early nucleation steps, ideally directly inside an unperturbed cell, is a challenge. The initial nuclei may be sub-diffractive in size (and indeed they are for this study), which makes them difficult to capture by conventional imaging techniques. In vitro imaging of de novo assembly must rely on high resolution techniques like scanning transmission electron microscopy (STEM), atomic force microscopy (AFM), or super-resolution fluorescence microscopy to observe early aggregates (*Gaczynska and Osmulski, 2008*; *Kaminski and Kaminski Schierle, 2016*; *Sunde and Blake, 1997*). Although super-resolution imaging has been performed in living cells after seeding the cells with preformed small fibers, these studies have focused primarily on capturing stably growing fibers (*Kaminski and Kaminski Schierle, 2016*) rather than the transient early precursors. To our knowledge the size distribution, cluster dynamics, and physical nature of early pre-nucleation stage protein aggregates have not been previously determined.

Here, we elucidate the early formation mechanism of misfolded protein aggregates directly in mammalian cells. We use super-resolution microscopy in fixed cells and light sheet microscopy in living cells to capture the nucleation stage of aggregate formation (*Figure 1A*). We find that precursor clusters exist in cells even under normal growth conditions. Moreover, the size distribution of these precursor clusters is robustly consistent with a textbook example of classical nucleation theory. Classical nucleation theory considers a simple question: Given a collection of molecules, if a subset of the molecules cluster together, would this raise or lower the energy of the system. It makes a specific prediction about the distribution of cluster sizes. An alternative mechanism, such as a vectorial active transport that locally clusters monomers, is unlikely to result in the same cluster size distribution.

The details of classical nucleation theory are revealing about how cells maintain homeostasis. The so-called homogenous nucleation is a prototypical mechanism by which first order phase transitions proceed (*Kalikmanov, 2013*; *Sear, 2007*; *Slezov, 2009*). A first-order phase transition describes the discontinuous changes needed for a system to go from a dispersed phase to a condensed phase (or vice versa). This may correspond to the concentration of a single component from its dispersed phase (for example condensation) or the demixing of some components from a multicomponent mixture (for example liquid-liquid phase separation). In these cases, there exists a saturation concentration above which the system transitions from dispersed to clustered phase – or from the mixed to the de-mixed phase. The saturation point then categorizes systems in two possible regimes, either sub-saturated or super-saturated. In the sub-saturated regime, the concentration of monomers is lower than the saturation concentration. The system can form clusters, but the clusters spontaneously lose molecules and dissolve. The sub-saturated state is a stable state in that the system remains in the dispersed or mixed phase. In contrast, in a super-saturated state – when the ambient concentration is above the saturation concentration - there exists a critical cluster size, below which clusters spontaneously dissolve, and above which clusters will spontaneously grow. The super-saturated state is metastable because clusters reaching a size above the critical size will grow at the expense of the dispersed phase. The condensed phase is favoured, and the concentration of monomers in the disperse phase –that is the super-saturation level of the system– gradually decreases as larger clusters absorb monomers from the ambient environment. The theory of first order phase transitions makes a distinct prediction for the distribution of the dissolving clusters which can be used to distinguish sub-saturated from super-saturated states.

Because cells under normal growth conditions do not show large growing clusters traced by Synphilin1, the naive hypothesis is that the cell is in a sub-saturated state, which is normally a steady state. However, we find that even under normal growth conditions, the distribution of clusters suggest the cells are in a super-saturated state, which is normally not a steady state. Rather, our results suggest that cells maintain homeostasis through a so-called *Szilard model* of non-equilibrium steady-state super-saturation (*Farkas, 1927*; *Slezov, 2009*). The Szilard model describes how a system can be maintained in steady state super-saturation if there is a mechanism to constantly clear the largest clusters. This size-dependent clearance of large aggregates appears to be mediated by the putative chaperone RuvbL.

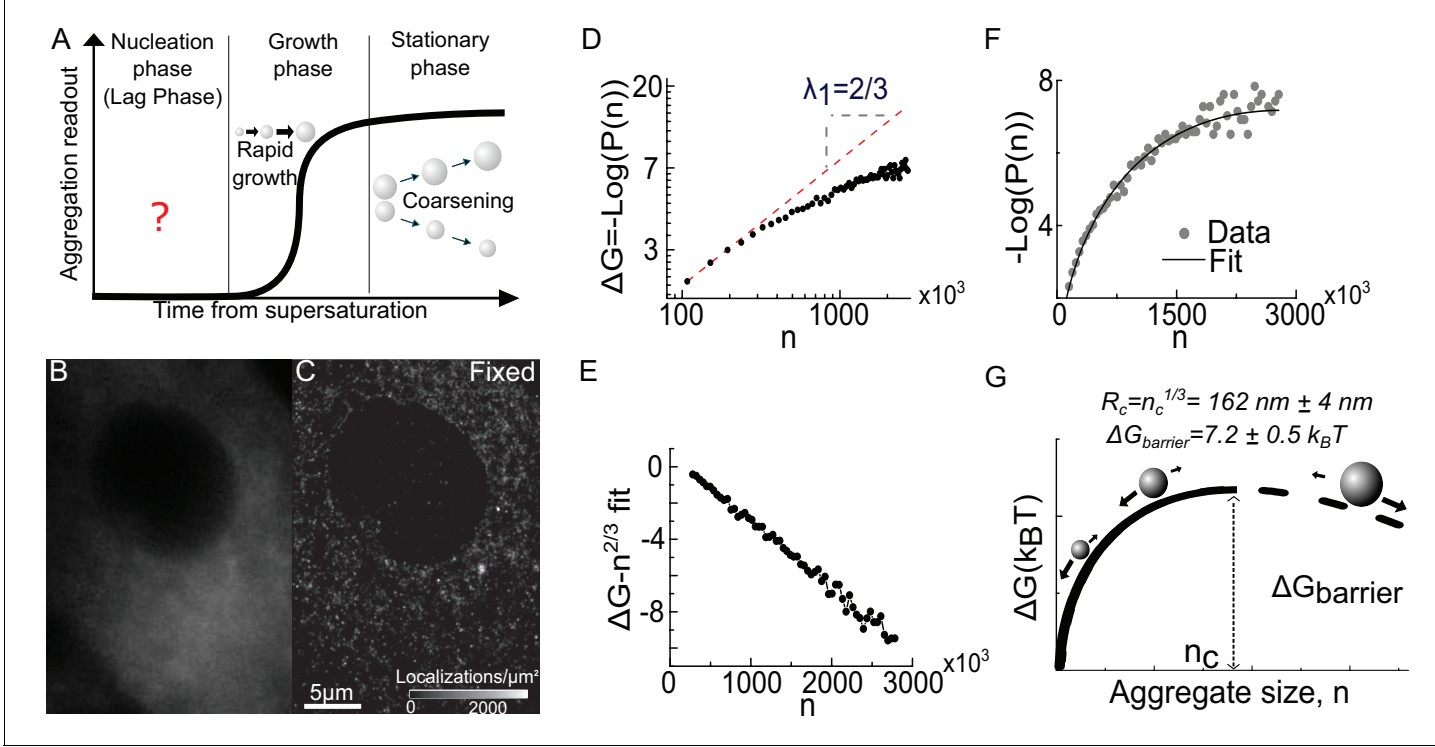

**Figure 1.** Super resolution imaging of fixed cells suggests that a condensation transition underlies aggregate formation. (**A**) Schematic of traditional experimental readouts of protein aggregation. Such measurements are blind to the nucleation phase but begin to show signal during the growth phase and stationary phase. (**B**) Conventional fluorescence image of Synphilin1-Dendra2 in mammalian cell. (**C**) Super resolved reconstruction of the same cell reveals multiple sub-diffractive aggregates (see *Figure 1—figure supplement 1* for representative DBSCAN images). (**D**) log-log plot of the free energy from the distribution of aggregate sizes (see *Box 1*) reveal a low-n asymptote of slope 2/3, reminiscent of a surface energy limited system ($an^{\frac{2}{3}}$) (See *Figure 1—figure supplement 3* for evidence for homogenous decoration of clusters by Synphilin1). (**E**) Plot of the resultant after correction of surface energy: by fitting an $n^{2/3}$ surface energy term from the asymptote of (D) and subtracting it from the data,the resultant data is strikingly linear, and has a negative slope, suggesting the second term ($-bn$). (**F**) Fit of the distribution of aggregate sizes to the full function $an^{\frac{2}{3}} - bn$ to obtain the a and b parameters. The parameters define the nucleation barrier height and critical aggregate size (see *Figure 1—figure supplement 2* for expression level control and alternative tracer control, see *Figure 1—figure supplement 5* for application to in neuronal precursor cells). (**G**) Schematic of the energy function, as predicted for a super-saturated system undergoing first order phase transition. The line is dashed after the critical size to highlight that the functional form is experimentally valid only below this size. The barrier height and critical radius from the fit in F are represented (see *Figure 1—figure supplement 4* for larger schematic and definition of terms in text). Data is from 25,000 aggregates from 28 fixed cells. Errors represent s.e.m. All cells imaged in this figure were fixed cells. Log refers to the natural logarithm (base 'e') (see *Figure 1—figure supplement 5* for control in neuronal precursor cell line).

DOI: https://doi.org/10.7554/eLife.39695.002

The following figure supplements are available for figure 1:

**Figure supplement 1.** Super-resolution imaging and cluster identification using DBSCAN.
DOI: https://doi.org/10.7554/eLife.39695.003

**Figure supplement 2.** The effect of expression level, alternative protein markers.
DOI: https://doi.org/10.7554/eLife.39695.004

**Figure supplement 3.** The clusters have a well defined density.
DOI: https://doi.org/10.7554/eLife.39695.005

**Figure supplement 4.** Definition of terms in condensation theory.
DOI: https://doi.org/10.7554/eLife.39695.006

**Figure supplement 5.** Comparison between cluster size distributions in MCF10A and Neuro2A cell lines.
DOI: https://doi.org/10.7554/eLife.39695.007

## Results

### Super-resolution imaging of fixed cells suggests classical nucleation theory underlies aggregate formation

We engineered mammalian cell lines expressing Synphilin1 - a tracer of aggregates in Parkinson's disease (*Chung et al., 2001*; *Tanaka et al., 2004*; *Wakabayashi et al., 2000*) - fused to a fluorescent protein Dendra2 (*Chudakov et al., 2007*). Dendra2 is a green to red photo-convertible protein that enables photo-activation localization microscopy (PALM) (*Betzig et al., 2006*), a single-molecule based super-resolution (*Betzig et al., 2006*; *Hess et al., 2006*; *Rust et al., 2006*) approach we used previously to study protein clustering in mammalian cells (*Cho et al., 2016*; *Cisse et al., 2013*). How Synphilin1 is recruited to aggregates is not fully understood. However, this protein is a commonly used tracer for well-studied misfolded protein aggregates such as Lewy bodies (*Tanaka et al., 2004*; *Wakabayashi et al., 2000*). Here, we concentrate on sub-diffractive Synphilin1 traced aggregates whose size distribution we measure. We checked that neither the expression level of Synphilin1 tracer protein nor the identity of the tracer (alternative tracer alpha-Synuclein) have any detectable effect on the size distribution of sub-diffractive clusters (*Figure 1—figure supplement 2*). This suggests that Synphilin1 in our sub-diffractive clusters merely serves as a tracer and does not on its own affect cluster formation at the expression levels tested.

Wide-field epi-illumination (conventional) imaging of Synphilin1 in a fixed cell showed a diffuse cytoplasmic signal without any apparent aggregation (*Figure 1B*) as expected for a normal (i.e. without drug treatments) cell. However, super-resolution imaging of the same cell clearly revealed a large population of sub-diffractive clusters (*Figure 1C*).

We characterized the properties of these sub-diffractive clusters using density based spatial clustering of applications with noise (DBSCAN) (*Ester et al., 1996*) (*Figure 1—figure supplement 1*). We measured the radius and the number of localization events (corresponding to the fluorescent photo-activation and detection events) (see Materials and methods and *Figure 1—figure supplement 3*). We find that the number of localization events in a cluster, scales with the cube of the measured cluster radius This suggest that, at the relevant cluster sizes, the fluorescent detection events of the Synphilin1 tracer protein may be spread throughout the cluster volume at uniform density (*Figure 1—figure supplement 3*). Only clusters with a radius greater than our localization accuracy [estimated to be ~20nm (*Cho et al., 2016*)] are interpreted in our analysis. For the analysis that follows, we defined the cluster size as a variable '$n$', given by $n = \left(\frac{R}{1\,nm}\right)^3$ where R is the measured cluster radius in nanometres (*Figure 1—figure supplement 3*). Here, the parameter $n$ is proportional to, but different from the actual number of molecules in a cluster; the proportionality constant is determined by the density of all monomers in the cluster which is not known.

Following our observation of sub-diffractive clusters in the cell, we searched for signs of a thermodynamically driven first order phase transition in which spontaneous nucleation and growth mechanisms arise (*Slezov, 2009*). In condensation, the free energy change accompanying the clustering of n molecules into a single condensate is: $G = an^{2/3} \pm bn$ [See *Box 1*]. The first term is the surface term and accounts for the interfacial energy of the $n^{2/3} \propto R^2$ molecules on the surface of the condensate; the parameter '$a$' serves as (positive) surface energy parameter. The second term is the bulk term and depends on the total number of molecules in the volume ($n^1 \propto R^3$) of the cluster; '$b$' serves as a (positive) bulk energy parameter, and the positive or negative signs in front depends on whether the system is sub-saturated or super-saturated respectively [See *Box 1*]. The theory of first order phase transitions allows the determination of the free energy directly from the distribution of cluster sizes (which will follow $P(n) = Ae^{-G/k_BT}$ but with corresponding positive or negative bulk energy term in the free energy differentiating between sub- or super- saturation respectively) [See *Box 1*]

To test the applicability of this theory to our data, we examine the distribution of cluster sizes. For either a sub-saturated or a super-saturated system the free energy as a function of cluster size is $\Delta G(n) = -k_BTLog(P(n))$ [for n<$n_c$, in the super-saturated case (*Slezov, 2009*)] where $k_B$ is the Boltzmann constant, $T$ the temperature (in Kelvin); P(n) represents the histogram (formally the normalized distribution function) of cluster sizes, and $n_c$ the size of maximum $\Delta G$ [*Figure 1—figure supplement 4* and (*Slezov, 2009*)]. That any of this is valid for the formation of endogenous aggregates inside living cells – complex, highly regulated, multicomponent entities forming in an intrinsically non-equilibrium environment in vivo – we could not a priori know. However, by investigating the features of

# Box 1. Determining the free energy from distribution of cluster sizes.

Here we briefly describe how the free energy change in assembling a cluster of n molecules from the disperse phase - introduced in the main text as consisting of a surface and a bulk term – depends on system composition, how it describes behavior above and below the saturation concentration and how we can directly compute this function from experimental measurements. The equation $\Delta G = an^{2/3} \pm bn$ presented in the main text, is a simplified equation that can be derived exactly– it consists of two terms.

1) The Surface term $\Delta G_{surface} = an^{2/3}$ represents the energetic cost of setting up an interface between the clusters and ambient solution (or equivalently the interface between two liquid phases of de-mixed components in phase separation). The prefactor a is related to the composition of the system. Consider a cluster of n molecules of volume $v_n = nv_1$ where $v_1$ is the average volume taken by 1 molecule in the clustered phase. For spherical clusters of constant density $v_1 = M/\rho N_A$ (M is molar mass, $\rho$ is density and $N_A$ is Avogadro's number) and thus:

$$\frac{4}{3}\pi R_n^3 = v_n$$

$$and \ 4\pi R_n^2 = A_n$$

Therefore,

$$R_n = \left(\frac{3v_1}{4\pi}\right)^{1/3} n^{1/3} \ and \ A_n = (36\pi)^{1/3} v_1^{2/3} n^{2/3}$$

If the energy per unit area of the interface is $\sigma$– then the surface energy term is

$\Delta G_{surface} = \sigma A_n = \sigma (36\pi)^{1/3} v_1^{2/3} n^{2/3}$

So that we recover the form $\Delta G_{surface} = a \ n^{2/3}$. The prefactor $a = \sigma(36\pi)^{1/3} v_1^{2/3}$ depends on the specific interactions and geometric details of the interface between the two phases.

2) The bulk term, $\Delta G_{bulk} = \pm bn$ represents the difference in free energy between a system with all molecules in the ambient phase, and a system with n molecules in the clustered phase. In this case, it can be shown (**Abraham, 1974**) that for a cluster of n molecules

$\Delta G_{bulk} = -\Delta\mu \ n$ with $\Delta\mu = k_B T Log\left(\frac{c_{amb}}{c_{sat}}\right)$. Where $c_{amb}$ is the ambient monomer concentration and $c_{sat}$ is the saturation concentration - the concentration that would be at equilibrium with the clustered phase. Log refers to the natural logarithm (base 'e'). $\Delta\mu$ depends on both the ambient concentration of monomers and the interactions of the monomers in the two phases and changes sign at $c_{amb} = c_{sat}$. $\Delta G_{bulk} = \pm bn$ with the sign depending on whether $c_{amb} < c_{sat}$ (positive sign) or $c_{amb} > c_{sat}$ (negative sign) Therefore, for a given set of interactions, the ambient concentration alone can control which of the two phases is favoured and by how much. The system is referred to as sub-saturated when $c_{amb} < c_{sat}$, and thus the bulk term $\Delta G_{bulk} = +bn$. On the other hand super-saturated systems, $c_{amb} > c_{sat}$, result in a bulk term $\Delta G_{bulk} = -bn$.

Therefore, if the system is sub-saturated $\Delta G = an^{2/3} + bn$, both the terms add so that the free energy continuously increases with increasing cluster size n. This energy function has no maximum, and therefore there is no nucleation barrier beyond which clusters stably grow. Any cluster formed is energetically costly, and bigger clusters are increasingly more costly. Clusters that form in sub-saturated state will be driven to dissipate. In a sub-saturated system, the distribution of cluster sizes is given by $P(n) = Ae^{-G/k_B T}$ (where A is a factor that normalizes the probability distribution), and the free energy $\Delta G(n) = -k_B T Log(P(n))$ (offset by a constant due to the normalization factor A, see Materials and methods).

On the other hand, if the system is super-saturated, $\Delta G = an^{2/3} - bn$ with the two terms in $\Delta G$ having opposite signs. This balance between a positive surface energy and a negative bulk

contribution leads to a maximum in the free energy. An energy barrier and the critical cluster size for the system are at that point where the free energy function is maximal: $n_C = \left(\frac{2a}{3b}\right)^3$ Below this critical size, clusters formation and growth is energetically costly (positive slope $\Delta G$), and such 'sub-critical' clusters are thermodynamically driven to dissolve. Above this critical size, cluster growth is energetically favoured (negative slope $\Delta G$; if a cluster grows to a size greater than the critical size, it will grow at the cost of the monomer pool (and hence reduce the ambient concentration). For the duration while a super-saturated concentration is maintained, the theory has a simple prediction for the size distribution of sub-critical clusters. The sub-critical cluster size distribution is a Boltzmann distribution (**Slezov, 2009**) (but with the negative sign of the bulk term in $\Delta G$ in contrast to the sub-saturation distribution). $P(n) = Ae^{-\Delta G/k_B T}$ (where A is a factor that normalizes the probability distribution) and $\Delta G(n) = -k_B T Log(P(n))$ (offset by a constant due to the normalization factor A, see Materials and methods).

Such super-saturated states are normally transient. Given the interactions and the ambient concentrations, the free energy change favors the addition of each molecule to a super-critical cluster, thereby continuously decreasing the ambient concentration until it reaches the saturation level. Therefore in unusual cases the super-saturated state could be maintained if super-saturated clusters were removed from the system, this is the principle behind the so-called Szilard model.

the experimentally measured $\Delta G(n) = -k_B T Log(P(n))$ we find the energetics of a condensing super-saturated system. Since classical nucleation theory makes predictions on the distribution of sub-critical clusters, we consider the behavior of our experimentally measured free energy function $\Delta G(n) = -k_B T Log(P(n))$ for low $n$ values(Log refers to the natural log (Base 'e')). The log-log plot of our experimentally measured $\Delta G(n)$ reveals a linear asymptote of slope $= \gamma_1 = 2/3$ for small $n$ values (**Figure 1D**). This evokes a system dominated by a surface energy ($n^{2/3} \propto R^2$) for small clusters. We then fitted the first few data points of $\Delta G(n)$, corresponding to the smallest clusters, to estimate a surface energy term $an^{2/3}$ and subtracted it off of the data ($\Delta G(n)-an^{2/3}$) to obtain the resultant after surface correction. The resultant was linear ($\gamma_2$=1) to within our experimental uncertainty suggestive of a bulk (volumetric, $n^1 \propto R^3$) energy contribution (**Figure 1E**). Moreover the slope is negative, indicating that the (negative) bulk energy term ($-bn$) minimizes the free energy. A positive surface tension balanced by negative bulk energy is the key signature of a super-saturated condensing system; the combination gives rise to a well-defined maximum free energy representing the free energy barrier for nucleation ($\Delta G(n_c)$), and the corresponding critical cluster size $n_C = \left(\frac{2a}{3b}\right)^3$ above which clusters are thermodynamically stable and will spontaneously grow. By contrast, a sub-saturated system has the same surface term ($an^{2/3}$), but a positive ($+bn$) bulk term such that there is no maximum to the free energy and therefore no regime for stable cluster growth (**Slezov, 2009**). Therefore, the analysis of the cluster size distribution in unstressed cells reveals the precise energetics of a super-saturated system undergoing first-order phase transition (**Abraham, 1974**; **Slezov, 2009**).

By fitting the data, plotted as -Log(P(n)) to the functional form $\Delta G(n) = an^{2/3} - bn$ (**Figure 1F**) (where for the case described in **Box 1**, $a = \left[(36\pi)^{1/3}\right]\sigma v_1^{2/3}/k_B T$ is the dimensionless surface energy, $b = Log\left(\frac{c_{amb}}{c_{sat}}\right)$ is the dimensionless bulk energy term and $\Delta G(n)$ is consequently in units of $k_B T$), we obtain the two parameters $a = 1.07 \times 10^{-3} (\pm 0.06 \times 10^{-3})$ and $b = 4.3 \times 10^{-6} (\pm 3 \times 10^{-7})$ (best fit, mean $\pm$ (s.e.m)) which determine the thermodynamic properties of the condensation process (**Figure 1G** and **Figure 1—figure supplement 4**). Using these parameters, we can now extract two important biophysical properties of the process: the critical radius and the nucleation barrier (see Materials and methods). The nucleation barrier is $\Delta G = 7.2 (\pm 0.5) k_B T$, and the critical radius (i.e. the radius above which clusters will spontaneously grow) is $R_c$=162 ($\pm 4$) nm (mean $\pm$ (s.e.m)) (Materials and methods).

Because the critical radius $R_c$ is below the optical diffraction limit, this explains why a super-resolution technique was required to measure it. However, we were surprised that the value was as big as it is, because it is much higher than what would be predicted if a few monomers were sufficient to

nucleate a stable cluster. Our results suggest that the initial clusters likely form through a condensation of weakly interacting monomers.

We tested if these conclusions were generalizable to another cell type by performing our assay on clusters traced with dendra2-synphilin in Neuro2A cells (neuronal precursor cells, *Figure 1—figure supplement 5*); we also tested labeling aggregates with an alternate tracer protein alpha Synuclein (*Figure 1—figure supplement 2*). In all these cases we arrived at the identical distribution, suggesting the physical mechanism for sub-diffractive Synphilin1- or alpha Synuclein-marked aggregation may apply to a range of mammalian cells.

In summary, we developed a high-resolution imaging assay to look at the distribution of aggregate (cluster) sizes in fixed cell snapshots. We exploited the theoretical premise that in the distribution of cluster sizes below the critical size (sub-critical), the distribution can be approximated as a Boltzmann distribution. From this distribution we get the free energy. The resulting free energy terms extracted from the data were a surface tension term balanced by a linear bulk term, exactly as expected for the condensation of a super-saturated system. The free energy calculation allows us to determine the nucleation energy barrier and the critical cluster size (above which the clusters grow stably) for the aggregation inside mammalian cells. We could also determine how these quantities change under varying conditions. An important conclusion in this section is that even under normal growth conditions mammalian cells are super-saturated.

## Super-saturation can be tuned by the levels of endogenous aggregating polypeptides

To test our model, we study how drug treatments affect its parameters. As noted before, the energetics of nucleation and growth depend on two parameters corresponding to a bulk term and a surface term. These parameters derive from the microscopic biochemical interactions within the system, and should depend on the concentration of aggregating proteins (which affects the bulk term) and on their energy of interactions.

A major constituent of Synphilin1-traced aggregates is believed to be endogenous misfolded polypeptides (*Zaarur et al., 2008*). We employed methods to pharmacologically increase or decrease the cellular concentrations of these predicted constituents and measured the effect on cluster parameters ($R_c$ and nucleation barrier). To decrease the concentration of aggregating polypeptides we partially inhibited translation with Rapamycin (*Jefferies et al., 1997*) which reduces the pool of newly synthesized misfolded, therefore aggregation prone, polypeptides (*Conn and Qian, 2013*; *Sherman and Qian, 2013*). We confirm the results with treatment by direct translation inhibitor Cycloheximide. By contrast, to increase the concentration of misfolded aggregating polypeptides, we incubated cells either with azetidine-2-carboxylic acid (AZC), a proline analogue that promotes misfolding of newly synthesized proteins (*Goldberg and St John, 1976*), or with the proteasome inhibitor MG132 that reduces degradation of misfolded-proteins (*Nawaz et al., 1999*).

In *Figure 2A–F* we find that Rapamycin increases the nucleation barrier and critical radius, consistent with expectation for reduced super-saturation. A similar effect is also observed in treatments with the translational inhibitor cycloheximide (*Figure 2—figure supplement 1*). In contrast, treatment with either MG132 or AZC decreases the nucleation barrier and critical radius, consistent with expectation for increased super-saturation. Therefore, over a range of complex pharmacological perturbations –with simple intuition of how the perturbation would affect the saturation– the measured effects match the expectation from classical nucleation theory.

In summary, in this section we tested the effect of different pharmacological perturbations. We also confirmed here that the expression of the labelled tracer (Synphilin1) is not the main driving force for clustering, consistent with the previous characterization that the tracer labels aggregates that are mostly composed of misfolded polypeptides (*JA, 2006*; *Meriin et al., 2012*; *Park et al., 2017*; *Tanaka et al., 2004*). We find that the theoretical model holds robustly under different drug treatments that globally affect protein quality control: a drug that reduces global protein synthesis, thereby reducing the concentration of misfolded polypeptides is found to reduce super-saturation (i.e. increase the nucleation barrier and $R_c$) while a drug that promotes misfolding of newly synthesized protein is found to increase the super-saturation (i.e. decreasing the nucleation barrier, and $R_c$).

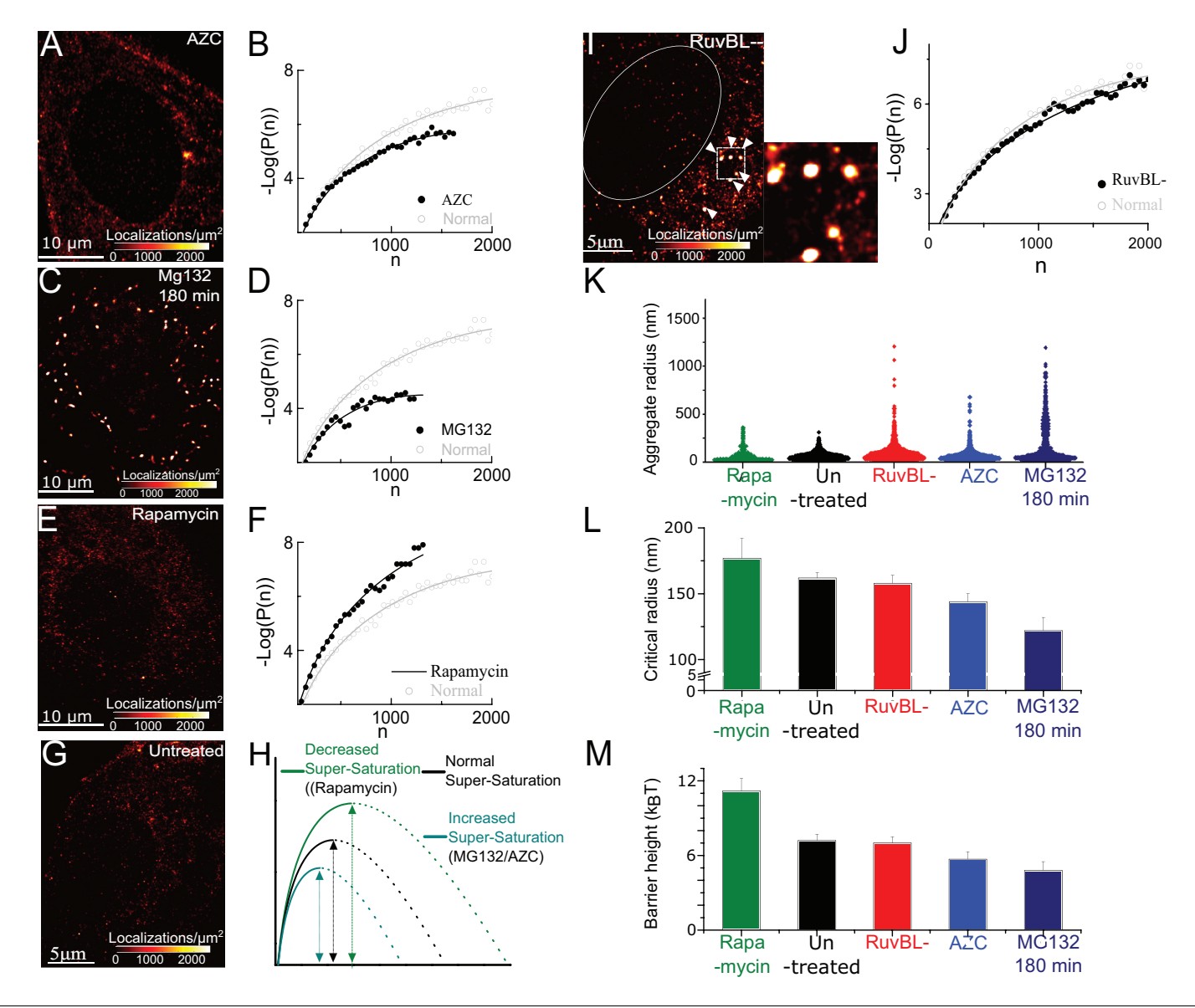

**Figure 2.** Super-saturation can be tuned by the levels of endogenous aggregative polypeptides and RuvBL-dependent mechanism clears super-critical clusters: (A–F) Representative super-resolved reconstruction and free energy functional fit for AZC- (**A and B**), MG132-(**C and D**), and Rapamycin- (**E and F**) treated cells. Distribution functions were computed from 2500 to 10,000 clusters from 7 to 10 cells in each condition. Red-hot colour code is used to indicate the relative density of detections in A, C, E. Gray plot in B, D, F represents the functional fit for untreated cells for comparison. (**G**) Representative super-resolution reconstruction for an untreated cell showing many sub-diffractive aggregates (dark red) but few large aggregates (red hot). (**H**) Schematic of the observed effects of the different pharmacological treatments. (**I**) Representative super-resolution reconstruction for a RuvBL depleted cell and zoomed view of large (red hot) aggregates. (**J**) –Log(P(n)) versus n curve from 7000 clusters from 9 RuvBL depletion cells shows almost no effect on the sub-critical distribution. (**K–M**), quantification of the effect of the perturbations and comparisons of relative changes in the distributions of aggregate size (violin plot, (**K**), $R_c$ (**L**) and the measure nucleation barrier; (**M**) the range of values in L and M is chosen to illustrate the main differences. Error bars in L and M represent s.e.m in fit estimation (Materials and methods). All cells imaged in this figure were fixed cells. Log refers to the natural log (base 'e') (See *Figure 2 – figure supplement 1* for combined effect of AZC and MG).

DOI: https://doi.org/10.7554/eLife.39695.008

The following figure supplement is available for figure 2:

**Figure supplement 1.** The effect of aggregation promoting amino acid substitute (AZC), proteasome inhibitor (MG132), translation inhibitor cycloheximide and the HSP70 inhibitor (Ver155008):
DOI: https://doi.org/10.7554/eLife.39695.009

## RUVBL-dependent mechanism clears super-critical clusters

The picture emerging from our analysis is that even untreated cells are super-saturated. In super-saturated systems if even one cluster reaches the critical size it should spontaneously grow; that cluster will come to dominate the system and absorb monomers until the system is no longer super-saturated. A violin plot of cluster sizes from untreated cells (*Figure 2K*, black) indicates that while a large population of small cluster sizes is apparent (indicated by the width of the violin plot in *Figure 2K*), there is a small minority of clusters (<5%) that have reached a size greater than $R_c$. Therefore, despite the fact that clusters do reach the critical size, such a population of super-critical clusters seems to be suppressed in healthy cells; this is corroborated by images of untreated cells that are distinctly devoid of large super-critical clusters (*Figure 2G*). In the theory of first-order phase transition, this observation is suggestive of a model with strict requirements: There ought to be a clearance mechanism that acts on the super-critical clusters without significantly affecting the sub-critical distribution.

Because a AAA+ ATPase, RuvBL, was previously suggested as a potential protein disaggregase in mammalian cells and in yeast (*Zaarur et al., 2015*), we tested whether RuvBL may be involved in the clearance of super-critical clusters. Consistent with this hypothesis, we find that knocking down RuvBL1 in untreated cells results in the appearance of large clusters (*Figure 2I*, compare to untreated cell *Figure 2G*). A violin plot of cluster sizes from RuvBL knocked-down cells shows a clear population of large cluster sizes. Some clusters have radii greater than $1\mu m$, a size range that we observed previously only after hours of proteasome inhibition (*Figure 2K*). These results implicate RuvBL in the clearance of large clusters from untreated cells (see *Figure 3* for further tests of RuvBL).

Importantly, we find that upon RuvBL knockdown, $R_c = 157 \pm 6\ nm$ did not change significantly from $R_c$ in control untreated cells ($162 \pm 4\ nm$) (*Figure 2J*) suggesting that RuvBL knockdown did not significantly change the sub-critical distribution. This observation implies that RuvbL did not affect the concentration of aggregating molecules or their interactions, unlike, for instance,

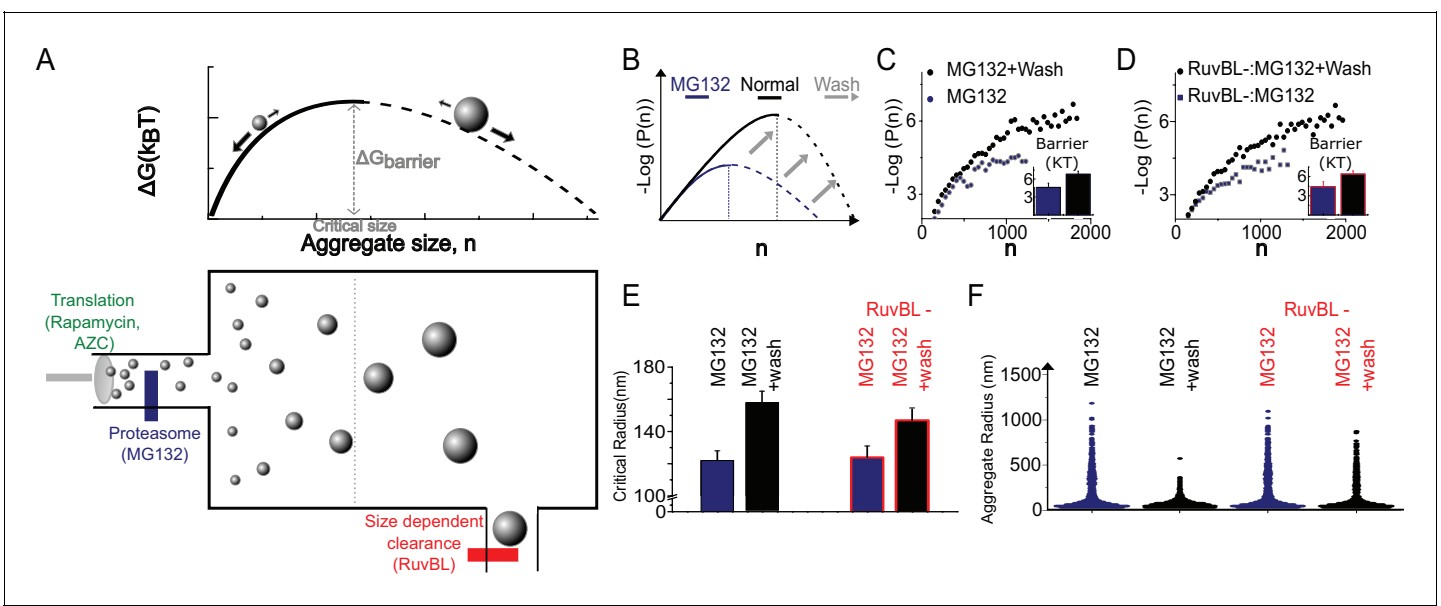

**Figure 3.** The Szilard non-equilibrium steady state accounts for all genetic and pharmacological and genetic stresses. (**A**) Schematic depiction of Szilard mechanism for maintenance of super-saturated steady state by simultaneous regulation of monomer creation and super-critical elimination. The suggested mode of action of the stress conditions utilized in *Figure 2* and the RUVBL depletion discussed in *Figure 2*, and *Figure 3* are indicated. (**B**) Schematic of predicted effect of Mg132 washout on sub-critical cluster size distribution and free energy function. (**C, D**) Data plots of ΔG (n) vs. *n* measured before and after washout both without (**C**) and with (**D**) RuvBL depletion. Each curve is generated from 2500 to 6000 clusters from 6 to 10 cells in the various conditions. (**E**) Critical radii measured with and without RuvBL depletion, before and after MG132 washout. (**F**) Violin plots of cluster sizes in the four conditions of (**C–E**) showing that the super-critical clearance is impeded in the case of RuvBL depletion compared to un-depleted cells. In (**B–D**), Log refers to the natural log (base 'e').

DOI: https://doi.org/10.7554/eLife.39695.010

proteasome inhibition, and AZC incubation which reduced $R_c$ (*Figure 2B, D, E*). Our data indicate that RuvBL-dependent clearance of clusters acts preferentially on clusters that have reached a size above $R_c$, without changing the sub-critical distribution or the nucleation process.

We also tested whether the classical HSP70-mediated pathway for disaggregation was involved at this stage. HSP70 is known to disaggregate amyloid fibrils (*Gao et al., 2015*) and the chemical Ver155008 inhibits HSP70 (*Massey et al., 2010*). However, here, using HSP70 inhibitor Ver155008 we were not able to generate a substantial population of super-critical clusters (*Figure 2—figure supplement 1*). This suggests that the RuvBL mechanism is likely distinct from HSP70 mediated disaggregation.

In summary, the ultimate end point of the phase transition that occurs in super-saturated systems should be the formation of a large macroscopic condensed phase. We found it very surprising that cells are super-saturated yet they do not have many super-critical clusters under normal growth conditions. We argue that there exists a mechanism that preferentially clears large aggregates without significantly affecting the pre-nucleated distribution of aggregates. We identify a putative protein chaperone (RuvBL) that is key to this clearance mechanism.

## The Szilard model for non-equilibrium steady state accounts for the genetic and pharmacological stresses tested

Without a specific mechanism in place to maintain the state, a super-saturated distribution is transient and unstable. Even if new monomers are continuously produced, super-critical clusters spontaneously absorb monomers faster than new clusters can form. One mechanism, attributed to Leo Szilard (*Farkas, 1927*; *Slezov, 2009*), was proposed to maintain a super-saturated distribution at steady state, through the preferential clearance of super-critical clusters.

The Szilard mechanism maintains a non-equilibrium steady state through the continuous production of aggregating monomers, their constant thermodynamically driven condensation (governed by the free energy function describe earlier) and a mechanism for preferential clearance of clusters at sizes greater than the critical size. Our data suggest a Szilard-type model (*Figure 3A*) whereby a RuvBL-dependent mechanism may help maintain cellular homeostasis in normal cells. Constituent monomers – the concentration of which may be affected through modulation of protein misfolding, damage, translation or degradation (e.g. through drug treatment or aberrant cellular processes) – condense through weak biochemical interactions with their condensation limited by their surface to volume ratio. At some point after the clusters reach a critical size beyond which their growth is favoured, they are subject to RuvBL-mediated clearance. These elements together explain the striking agreement between the distribution of sub-critical clusters and the energetics of steady state super-saturation in first order phase transitions. It also accounts for the measured effects of pharmacological treatments (e.g. in *Figure 2*).

One implication of the Szilard model is that accumulation of super-critical clusters could result from two independent mechanisms: increase in super-saturation (for example by proteasome inhibition or by increased expression of polypeptides) or alternatively by decreasing clearance (e.g. RuvBL knock down). To test this inter-dependence, we performed the treatment by proteasome inhibitor MG132 followed by washout, in cells with or without RuvBL knockdown. In all cases, after MG132 washout the free energy barrier returned to close to that of normal, untreated cells, suggesting that regardless of RuvBL knockdown, MG132 acted to increase the super-saturation in a reversible manner (*Figure 3C–E*). However, after washout the clearance of the super-critical clusters (which existed during MG132 treatment) was only possible in the controls with normal levels of RuvBL. On the other hand, in the cells where RuvBL was knocked down, although the free energy barrier returned close to normal untreated level, a population of super-critical clusters remained after MG132 wash-out, corroborating the fact that RuvBL was necessary for the clearance of the super-critical clusters. These experiments lend further support for RuvBL's role as a part of a super-critical cluster clearance mechanism and help to demonstrate how cellular relaxation and adaptation after complex combinations of perturbations can be fully explained by the Szilard steady state model.

## Live cell imaging data further support the model of super-critical clusters as condensates

All the analysis and conclusions thus far were driven by a free energy function extracted from a distribution of cluster sizes in fixed cell snapshots. This model has implications about the clusters, which can be tested in living cells. First, we aimed to test whether the same free energy function describes the distribution of clusters in live cells. This required the development of a light-sheet-based imaging assay where the brightness of the cluster is used as a proxy for cluster size. The distribution obtained by light sheet imaging is biased toward brighter (i.e. larger sized) clusters compared to super-resolution imaging. Nonetheless we are able to resolve enough of the sub-critical distribution and verify that the same free energy function describes the live cell data. Secondly, we used this live cell data to estimate the surface tension of the cluster-cytoplasm interface. We found that the surface tension is comparable to previous estimates of phase-separated condensates in the literature. Third, we measured the growth kinetics of the few super-critical clusters found in the living cells. We found that the growth and behavior of the super-critical clusters was entirely consistent with the model – suggested by our analysis in the sub-critical regime- of cluster formation by condensation. In particular, we found that growth and shrinkage was size dependent as expected from classical nucleation theory. We also found clusters underwent merger events (~1 mergers visible per cell imaged over ten minutes) as would be expected of condensates.

## Cluster size quantification in living cells corroborate fixed cell data

To study the nucleation process directly in living cells, the relatively fast motion of clusters in living cells precluded our previously developed quantitative live cell super-resolution approaches (*Cho et al., 2016*; *Cisse et al., 2013*). Here, we develop a complementary, light sheet based, approach to study aggregation directly in the living cells. We opted for a non-diffractive, lattice light sheet (*Chen et al., 2014*) approach to selectively illuminate a thin optical sheet in the living cells; the reduced background of excited molecules we anticipated would give us sufficient contrast to detect sub-diffractive clusters as bright diffraction-limited spots.

We found that the light sheet reveals many condensates as diffraction-limited spots throughout the cell (*Figure 4A* and *Videos 1* and *2*). Since light-sheet microscopy is not inherently a super-resolution technique, the apparent spot size is not the physical size of the cluster and it is impossible to accurately correlate the total intensity of the clusters with their real volume. However, since we have

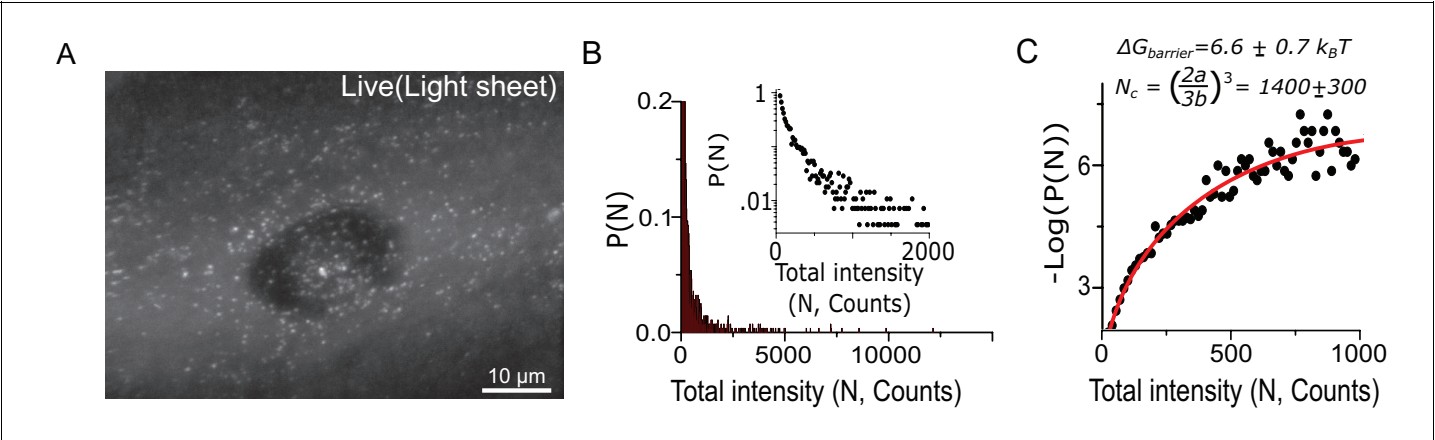

**Figure 4.** Cluster size quantification in living cells corroborates fixed cell data and the estimation of surface tension at the condensate-cytoplasm interface. (A) 2D maximum intensity projection of 3D direct imaging of Dendra2-Synphilin1 traced aggregates in a representative cell imaged with light sheet microscope. (B) Using the relative intensities rather than the radius as a measure of aggregate size, we may plot distributions that reproduce all the features of the fixed cell measurements (N=2800 aggregates from 22 cells). Insets: log-linear plot. (C) The free energy functional fit (yielding $a = 0.166 \pm .01$ *and* $b = 0.011 \pm .001$(best fit $\pm$ s.e.m)) of the live cell data corroborates the conclusions of the condensation through first order phase transition. Here, the variable parameter N is an estimate of the number of fluorescent Dendra2-Synphilin imaged per aggregate and Log refers to the natural log (base 'e'). All cells imaged in this figure were live cells.
DOI: https://doi.org/10.7554/eLife.39695.011

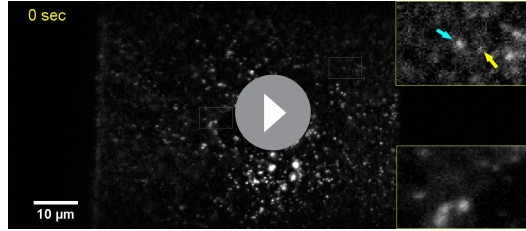

**Video 1.** Lattice light sheet imaging of Synphilin cluster mergers in live cells - example 1.
DOI: https://doi.org/10.7554/eLife.39695.012

**Video 2.** Lattice light sheet imaging of Synphilin cluster mergers in live cells - example 2.
DOI: https://doi.org/10.7554/eLife.39695.013

established from fixed cell measurements that out tracer protein Sypnhilin1 is homogenously distributed throughout the volume of clusters over the range of cluster sizes (*Figure 1—figure supplement 3*), the total fluorescence intensity of the directly illuminated, pre-converted (green state) Dendra2-Synphilin1 signal can serve as an estimate of the number of fluorescent molecules.

We obtain a live cell cluster size distribution (*Figure 4B and C*) that corroborates our conclusions. The distribution fits well to the functional form $\Delta G(N) = aN^{2/3} - bN$, and the measured nucleation barrier in live cell is ~6.6 (±0.7) $k_BT$ in agreement with the 7.2 (±0.5) $k_BT$ barrier measured by fixed cell super-resolution in *Figure 1*.

The critical cluster is estimated to have $N_c$ = 1400 (±300) labeled, fluorescent molecules in a live cell. Taken together with the super-resolution measured $R_c$, this suggests a spacing of ($N_c/R_c$) at most one Dendra2-Synphilin every 15 nm. Therefore, given a Stokes radius of Dendra2 ~1.5 nm, the labeled molecules occupy only about a thousandth of the volume of a cluster. This supports further that the clusters are likely heterogeneous and composed of far more unlabeled endogenous proteins than the fluorescent tracer (Synphilin 1) that labels aggregates.

## Estimation of surface tension at the condensate-cytoplasm interface

The live cell measurements allow us to make bounded estimates for the surface tension of the condensate-cytoplasm interface. Estimates for the surface tension allow for direct comparison between condensates formed in a wide range of systems, and in different studies. Briefly, the measured value of $a$ plays the role of a surface tension in the theory, $a^* = \left(\frac{r_1^2 \sigma}{k_b T}\right)$, with $r_1$ the length-scale of a monomer (taken to be ~1nm), $\sigma$ the surface tension; we previously noted that our definition of $n$ in fixed cells (or N in live cells) were correct up to a multiplicative constant. While this does not affect our ability to calculate the critical radius and nucleation barrier (see materials and methods), it does impact our estimate of the individual parameter $a$. Nonetheless, we can still provide bounds on the value of the surface tension at the condensate-cytoplasm interface.

In the fixed cell data we measure the radius R with high spatial resolution and then calculate $R^3$ but $n_{tot} = R^3 \rho$ where $\rho$ is the density of polypeptides in the cluster which is an unknown. In the live cell measurements we measure the total intensity I and calculate $N = I/I_{Single\ mol}$ but again $n_{tot} = Nk$ with k unknown. So both $N = I/I_{Single\ mol} = n_{tot}/k$ and $n = R^3 = n_{tot}/\rho$ are only measures of the total number of molecules up to a proportionality factor. Therefore from our fits we in fact get $a = a^*\rho^{2/3}$ *and* $b = b^*\rho$ from the fixed cell data $a = a^*k^{2/3}$ *and* $b = b^*k$ from live cell data - where starred parameters represent the real energetic terms if we had measured the total number of molecules. Since surface tension is determined by $a^* = \left(\frac{r_1^2 \sigma}{k_b T}\right)$, and we do not know $a^*$, we need to use bounds on $\rho$, k in order to estimate the surface tension.

For upper bounds - from live cell data we know that $n_{tot}$ is greater than the number of molecules that are fluorescing (only Synphilin1-Dendra2). That is, the real parameter $a^*$ is less than the measured parameter $a$. Consequently the surface tension calculated from the measurement over-estimates the real surface tension and is an upper bound – since the measured parameter $a$ (from live cell data) was 0.166 at 310K (assuming ambient temperature as the temperature) and using $a^* = \left(\frac{r_1^2 \sigma}{k_b T}\right)$, this works out to $\sigma < 6 \times 10^{-4} N/m$. From fixed cell data, taking as input from the live cell measurements that a critical cluster has at least 1400 molecules (the number of Synphilin

molecules estimated by intensity measurements), and that the critical cluster size is measured to be 162 nm, we can estimate a lower bound on the density. That density is at least $3.3 \times 10^{-4}$ *molecules per* $nm^3$. Since the measured value of $a$ was 0.001 we can use $a = a^* \rho^{2/3}$ to get an upper bound on the real value of $a$ and consequently on the surface tension this works out to $\sigma < 8 \times 10^{-4} N/m$.

Lower bounds may be obtained by noting that the density of the cluster cannot be more than 1 molecule per nm$^3$. This value corresponds to the size of proteins involved in the process. Using this value of $\rho = 1$ *molecule per* $nm^3$ we can compute the value of $a^*$ to be at least the measured value $a$ ($a$= 0.001 in fixed cells) and consequently establish a lower bound of $\sigma > 4 \times 10^{-6} \frac{N}{m}$.

Interestingly, the values $\sigma \sim 10^{-6}$ *to* $10^{-4} N/m$ are similar to the order-of-magnitude estimate for the surface tension of in vivo liquid droplets more apparent in large oocytes and embryos. For example a surface tension around $10^{-5} N/m$ was estimated for the nucleolar interface in oocytes (*Brangwynne et al., 2011*) or $10^{-6} N/m$ for germline P-granules liquid droplets interface with the cytoplasm in *c. elegans* (*Brangwynne et al., 2009*). The comparable values of surface tensions suggest that despite differences in their sizes, our diffraction-sized clusters may have the same droplet-like properties as previously reported large in vivo condensates. It is also feasible that cluster formation mechanism described in this study may be in play in the formation of larger in vivo condensates reported in other studies.

## Live cell cluster dynamics reveal key signatures associated with condensing systems

Nucleation and growth describes the evolution of a system toward matured, stable clusters. In our live cell imaging, we expected most of the clusters to be unstable because they were either sub-critical or under the influence of RuvBL clearance. Consistent with this expectation, comparable to fixed cells, we find that less than eight percent (<8%) of clusters were in the super-critical size range needed for stability, and most of the tracked clusters did not last as long as the full duration of our live cell experiments (6 min). Of the few that survived, we measured the dynamics of a representative population: we investigated the growth and shrinkage dynamics of 30 individual clusters from seven living cells imaged with the light sheet at 15 s time interval over 6 min. These represent a pool of the largest clusters; smaller clusters than this pool did not last the full 6 min of imaging or could not be tracked.

First, by normalizing the cluster intensity at t = 0, we find that individual clusters have gradually grown or shrunk by up to 40% in the course of 6 min (*Figure 5A*). We color-coded the growth/shrinkage dynamics with clusters growing more than 10% in red, and those shrinking more than 10% in blue. Those clusters that have not changed in size by more than 10% after 6 min are in black. Strikingly, the growth/shrinkage correlates directly with the original cluster intensity: *Figure 5B* shows that growing clusters (red) were almost unanimously the largest clusters in the pool, while the shrinking clusters (blue) were the smallest. The unchanged clusters had intensities N ~ 2000 close to the estimated $N_c$, indicating that these stable clusters are likely around or above the critical size.

This coarsening behavior, that large clusters tend to grow larger while smaller clusters tend to shrink, is a key dynamic signature of complex systems in phase transition, an example of which is the well-known phenomenon of Ostwald ripening (*Ostwald, 1897*) where larger clusters, owing to the lower curvature at their surface have a reduced pressure differential (Laplace pressure $\Delta P = 2\sigma/r$) across their surface and consequently reduced solubility relative to smaller clusters. Within the framework illustrated in *Figure 1* and *Figure 4*, coarsening can be understood by the fact that smaller condensates have a higher surface to volume ratio and therefore a higher free energy than large condensates; as such, above the critical barrier, large clusters spontaneously grow larger, and in steady state this happens at the detriment of smaller aggregates which will shrink.

Instantaneously, any given cluster can exhibit growth and shrinkage steps. And a few small aggregates will stochastically grow to reach super-critical sizes. Consistent with this view, in the time traces of *Figure 5B* one of the smaller aggregates has grown (red) over the imaging window. These reversible steps are consistent with a thermodynamically driven, probabilistic process.

Over the course of imaging, the overall growth or shrinkage seems roughly linear with time (N(t) $\propto$ t) for the individual clusters. We note that since our super-resolution data shows that cluster size scales with the cube of the average radius (*Figure 1—figure supplement 3*), our live cell data is

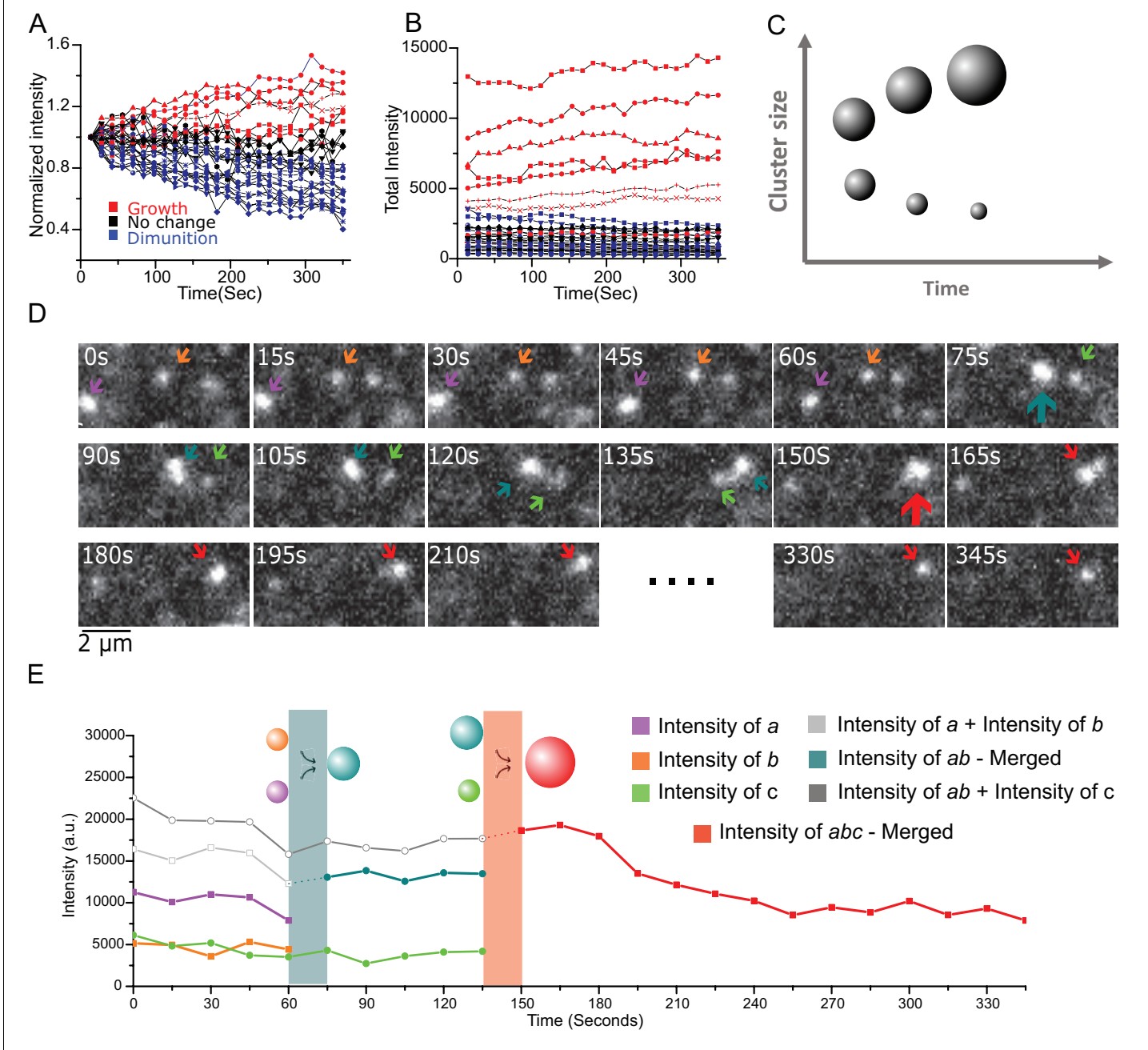

**Figure 5.** Live cell imaging of single aggregates growth and shrinking dynamics show ripening and coalescence. (A) Aggregate growth and diminution kinetics over a period of 6 min from 30 long-lived aggregates from seven cells. Here, the intensity is normalized to the initial intensity. Color code: red are growing intensity, blue are diminishing intensity, and black represent no change (<10%) between start and end time points. (B) The intensity time traces of the individual aggregates show a clear difference in the global trend with the initially larger aggregates also corresponding to the growing (red) aggregates, and initially smaller aggregates shrinking (blue). Instantaneously all aggregates exhibit individual steps of both growth and diminution, and one small aggregate out of the 30, was seen to grow more than 10%. (C) Schematic depicting the expectations from Ostwald ripening: This coarsening/ripening behavior that large aggregates grow larger while small aggregates diminish is a key dynamic signature of a fluid in phase transition. (D and E) The aggregates coalesce. (D) A montage three aggregates successively coalescing into one aggregate over the period of ~6 min. (E) The intensity trace of the three clusters in (D) shows that in each time upon merging the new droplet is the sum of the two precursor aggregates. Also see *Figure 5*; *Video 1* associated with 4D and 4E and *Figure 5*; *Video 2* for more examples). All cells imaged in this figure were live cells.
DOI: https://doi.org/10.7554/eLife.39695.014

consistent with N(t) $\propto$ R$^3$ $\propto$ t. Therefore, our observations of individual cluster growth and shrinkage in live cell would be consistent with condensed phase coarsening with R $\propto$ t$^{1/3}$, the prediction for both diffusion-limited Ostwald ripening, or coarsening by Brownian motion-induced coalescence (*Berry et al., 2015*; *Ratke and Voorhees, 2002*; *Slezov, 2009*).

In addition, we observe in the living cells that diffusing clusters, upon contact, merge into a single condensate (*Figure 5D and E* and *Videos 1* and *2*). In 15 cells imaged for 6 min each, we manually identified nine merger events, suggesting a rate of occurrence of about 1 in every 10 min. This coalescence is another common feature of systems undergoing condensation, where the net attractive interaction between monomers and the fact that the surface energy cost is minimized by fusion favor coalescence upon contact. Together, the observations from intensity distributions in live cells that is the corroboration of a thermodynamic free energy akin to condensation, the estimation of a surface tension comparable to phase separated biomolecules in vivo (*Brangwynne et al., 2009*; *Brangwynne et al., 2011*), the coexistence of fast stochastic dynamics with slow size-dependent coarsening, and the incidents of coalescence – all provide support for the conclusion that aggregate formation and growth is governed by the thermodynamics of a first order phase transition in live mammalian cells.

## Discussion

Aggregation of misfolded proteins is typically understood as sequential oligomerization in discrete, often irreversible steps adding monomers to an already formed nucleus (*Serio et al., 2000*). However, how the original nucleus forms in live cells has not been clarified. Our work establishes that a first-order phase transition is an appropriate thermodynamic framework to describe de novo formation and growth of protein aggregates in mammalian cells. Our results reveal that at the early stages, the formation of misfolded protein clusters is akin to a first order phase transition leading to the formation of a condensate. This link between condensation and early steps of aggregation provides a conceptual framework for our understanding of aggregation in vivo, with clear predictions and implications that can be tested and falsified in experimental studies.

A key prediction from the existence of the free energy form is that super-saturation is being maintained in steady-state in the living mammalian cells. One testable implication is that even healthy cells are already super-saturated and may have sub-diffractive condensates that may be impossible to detect by conventional imaging, but that are readily apparent with the high-resolution microscopes used here.

A requirement for physically maintaining a super-saturated steady state is that a mechanism exists to recognize and clear condensates that are larger than a critical size. This expectation has led us to identify the RuvBL-dependent pathway for aggregate clearance. It is possible that an accumulation of super-critical clusters could result from the misregulation of RuvBL-dependent clearance of super-critical clusters. We found that knocking down RuvBL was sufficient to increase super-critical clusters in the cells. In future studies, it would be interesting to look for correlations between disease onset and RuvBL expression level, or mutations.

Similarly, it has been hypothesized that intermediate aggregates could be more toxic to cells than late stage aggregates in aggregation diseases (*Cookson, 2005*; *Gosavi et al., 2002*; *Karpinar et al., 2009*; *Lashuel et al., 2002a*; *Lashuel et al., 2002b*; *Pountney et al., 2004*; *Ross and Poirier, 2004*; *Xu et al., 2002*). If maturing condensates represent the toxic species, then it would help explain the need for maintaining homeostasis by clearing condensates above a critical size. The Szilard model offers a predictive quantitative framework for testing such hypotheses, since it predicts and describes how alterations in cellular protein homeostasis would thermodynamically alter the balance between condensate formation and clearance.

Most aggregates studied in vitro require higher than physiological concentrations of proteins for de novo nucleation and formation of plaques, fibers or inclusions of homogeneous compositions (e.g. with a single protein constituent). Often, however, the protein's expression is not increased in cells that contain visible aggregates in vivo, raising a question on whether there are missing nucleators that facilitate this transition. Rather, in vivo condensates have very high concentrations of constituents, andcondensates can mature and acquire a variety of complex physical properties (including gelation or solidification [*Patel et al., 2015*] and irreversibility) over long time scales.

There is mounting evidence in the literature suggesting that in disease, intermediate steps of aggregate formation may proceed through the type of nucleation process that we observe here. In a classic study (*Serio et al., 2000*), Lindquist and colleagues examined the in vitro mechanism by which a prion protein Sup35 forms self-seeded amyloid fibers. They found that variably sized spherical complexes (akin to condensates and apparent by high-resolution imaging (STEM and AFM)), appear to be crucial intermediates in de novo amyloid nucleation (*Serio et al., 2000*). Similarly for Huntingtin N terminal fragments, in vitro studies (*Crick et al., 2013*; *Posey et al., 2018*) show that below the concentration required for growth of a structured fibrilar phase, there exists another saturation point distinguishing monomers and small oligomers of <10 nm size from spherical complexes of ~25 nm size (*Posey et al., 2018*); globular complexes have also been isolated from Huntingtin rat models (*Sathasivam et al., 2010*). Intermediate steps are also suggested by fluorescence correlation spectroscopy studies of nucleation of amyloids, where hints of a second nucleation barrier between small multimers and large structured aggregates were seen (*Garai et al., 2008*).

In light of such precedence and the high concentrations needed for the nucleation (*Törnquist et al., 2018*) of fibrils in vitro, it is plausible that, the micro-environment within in vivo condensates such as the one found in our current study, could serve to provide the high concentrations needed for a secondary nucleation (*Buell et al., 2014*; *Buell et al., 2014*) of fibers or plaques with various properties (*Vitalis and Pappu, 2011*). Further, the regulatory template of the Szilard mechanism - with a preferential regulation of super-critical clusters – could be useful in re-examining previous results, such as for example, the regulated competition between fibrilar and amorphous aggregation of Amyloid beta (*Garai et al., 2018*). It is also possible that mechanisms similar to those described here may be relevant in other contexts (*Cho et al., 2018*; *Chong et al., 2018*; *Sabari et al., 2018*).

Our assays and analytical framework also highlight various avenues where progress in the experimental state-of-the-art in the near future may yield rich insights. On one hand, the super-resolution methods are still limited by their localization accuracy (in our case ~20 nm); if the critical cluster size is not greater, the sub-critical cluster sizes would not be readily quantifiable. Super-resolution techniques are also limited in temporal resolution, making them unsuited for imaging fast moving sub-diffractive clusters in living cell. On the other hand, the lattice light sheet approach we used for live cell imaging is not sensitive enough to study detailed dynamics of the smallest clusters. And the gradual photo-bleaching of the fluorescent labels limits how long individual clusters can be imaged in living cells. Given these challenges there are still many questions made addressable by the ability to apply thermodynamic principles to aggregation in living cells.

While our investigation has focused on aggregates related to Parkinson's disease, we note that the methodology can be readily extended to any protein that can be fluorescently tagged (for example fused to the GFP-like Dendra2). Similar mechanisms may in fact apply in other cellular processes. For example, there are large membraneless cellular organelles that have been shown to behave like liquids, suggesting that their formation is akin to liquid-liquid phase separation. Applying the approaches described in this paper would further define the biophysical properties underlying membraneless organelle formation and also reveal regulatory mechanisms.

# Materials and methods

**Key resources table**

| Reagent type (species) or resource | Designation | Source or reference | Identifiers | Additional information |
|---|---|---|---|---|
| Cell line (*H. sapiens*) | MCF10A | ATCC | ATCC:CRL10317 RRID: CVCL_0598 | |
| Cell line (*H. sapiens*) | Neuro2A | ATCC | ATCC:CCL131 RRID: CVCL_0470 | |

*Continued on next page*

*Continued*

| Reagent type (species) or resource | Designation | Source or reference | Identifiers | Additional information |
|---|---|---|---|---|
| Transfected construct | Synphilin-GFP | *Zaarur et al., 2008* | n/a | Generated by Sherman lab - published *Zaarur et al., 2008*, backbone pCXsbr |
| Transfected construct | Human Alpha Synuclein | Addgene | 51437 | |
| Transfected construct | Dendra 2 | Clonetech USA | PDendra2C | |
| Transfected construct | Synphilin-Dendra2 | This Paper | n/a | Generated from synphilin-gfp and dendra two plasmid above. Available from Cisse lab |
| Transfected construct | alpha Synuclein Dendra2 | This paper | n/a | Generated from human alpha Synuclein and Dendra two plasmid above. Available from Cisse lab |
| Sequence-based reagent | siGENOME Non-Targeting siRNA #5 | Dharmacon | D-001210–05 | |
| Sequence-based reagent | siGENOME RUVBL1 siRNA | Dharmacon | (D-008977–04) | |
| Chemical compound, drug | | | | |
| Chemical compound, drug | MG132 | Sigma Aldrich | M8699 | |
| Chemical compound, drug | Rapamycin | Sigma Aldrich | R8781 | |
| Chemical compound, drug | Cycloheximide | Sigma Aldrich | C7698 | |
| Chemical compound, drug | Azetidine-2-Carboxylic acid | Sigma Aldrich | A0760 | |
| Chemical compound, drug | Cholera Toxin | Sigma Aldrich | C8052 | |
| Chemical compound, drug | Human Insulin | Sigma Aldrich | I9278 | |
| Chemical compound, drug | Leboqitz L15 medium | Sigma Aldrich | 11415064 | |
| Chemical compound, drug | Hydrocortisone | Sigma Aldrich | H0888 | |
| Chemical compound, drug | dbCAMP | Sigma Aldrich | D0627 | |
| Chemical compound, drug | Ver155008 | Sigma Aldrich | SML0271 | |
| Chemical compound, drug | Lipofectamine RNAiMAX | Thermo Fisher | 13778030 | |
| Chemical compound, drug | Xtremegene 9 | Sigma Aldrich | 6365779001 | |
| Software, algorithm | qSR | *Andrews et al. (2018)* | | Software made available on repository - http://github.com/cisselab/qSR/ |

*Continued on next page*

*Continued*

| Reagent type (species) or resource | Designation | Source or reference | Identifiers | Additional information |
|---|---|---|---|---|
| Software, algorithm | Lattice Light sheet processing code | *Chen et al., 2014* | | |
| Software, algorithm | Multiple Target tracking(MTT) | *Sergé et al., 2008* | | |

## Contact for reagent and resource sharing

'Further information and requests for resources and reagents should be directed to and will be fulfilled by the Lead Contact, Ibrahim Cisse (icisse@mit.edu)

## Experimental model and subject details

Cell lines were generated from MCF-10A (human breast epithelial) cells grown in 50:50 DMEM/F-12 medium supplemented with 5% horse serum, 20 ng/ml epidermal growth factor, 0.5 µg/ml hydrocortisone (Sigma Aldrich), 10 µg/ml human insulin (Sigma Aldrich), and 100 ng/ml cholera toxin (Sigma Aldrich). In all cases, medium was supplemented with Penicillin-Streptomycin and incubated at 37°C in an atmosphere of 5% $CO_2$ in a water-saturated atmosphere.

For the Synphilin-Dendra2 cell line – The retroviral expression construct with C-terminally tagged Synphilin one sub cloned into pCXbsr vector described previously (*Zaarur et al., 2008*) was used with EGFP at the C-terminus of the construct replaced with Dendra2, PCR cloned from the P-Dendra2C plasmid purchased from Clontech, USA.

For the alpha Synuclein–Dendra2 cell lines – alpha Synuclein gene was copied by PCR from Addgene plasmid #51437 and cloned into P-Dendra2C plasmid purchased from Clonetech, USA. For MCF10A –alpha Synuclein cell line – alpha Synuclein-Dendra2 plasmid, was transiently transfected using extremegene9 and selected using Kanamycin resistance cassette in MCF10A (ATCC, USA) cells. The cell line was tested for mycoplasma contamination by the high-throughput sequencing facility at the Koch Institute using the Lonza MycoAlert Plus kit. The cell line tested negative for mycoplasma contamination. Cell line identity was authenticated by ATCC using STR profiling, and gave a 94% match to ATCC cell line CRL-10317(MCF10A). For Neuro2A- alpha Synuclein-Dendra2 control – alpha Synuclein-dendra2 plasmid, described above was transiently transfected using extremegene9 and selected using Kanamycin resistance cassette in Mouse neuroblastoma cell line - Neuro2A (CCL-131, ATCC, USA)

Neuro2A was maintained in culture in growth medium consisting of 45% of DMEM high glucose medium w/L-Glutamine (GIBCO, USA), 45% of OptiMEM1 medium (GIBCO, USA) and 10% of Fetal bovine serum (GIBCO, USA) Supplemented by Penicillin-Streptomycin and incubated at 37C in an atmosphere of 5% $CO_2$ in a water saturated atmosphere.

Differentiation to neuronal state was achieved by simultaneous lowering of serum content to 1% and addition of 0.5 mM dbcAMP (Sigma Aldrich, USA) as suggested in ATCC product manuals. In 2 days of growth, distinctly neuronal morphology was established in ~75% of the cells in culture and alpha Synuclein expression was observed throughout the cell but was enhanced at the tips of the finger-like processes of the cell. The neuronal state of N2A cells under dbcAMP differentiation has been established in the literature (*Tremblay et al., 2010*)

## Method details

### Genetic and pharmacological treatments

The various pharmacological and genetic stresses were applied as follows-

Naïve (unstressed) growth – cells grown in the culture medium described above were imaged without any stress. This condition was measured as control along with every other stress and on its own on three separate occasions (independently cultured and plated imaging dishes).

Proteasome inhibition – cells were incubated for 0 to 4 hr using 2 µM MG132 (Sigma Aldrich, USA) in normal growth medium. Additional tests were done with lower concentrations and slightly

longer incubation times, 500 nM MG132 was the lowest attempted concentration that allowed aggresome formation. This stress was measured on three separate occasions (independently cultured and plated imaging dishes).

Rapamycin incubation: Cells were incubated for 12 hr in normal growth medium supplemented with 100 nM rapamycin (Sigma Aldrich, USA) after being plated on the imaging coverslip. This stress was measured on two separate occasions (independently cultured and plated imaging dishes).

Cycloheximide incubation: Cells were incubated in normal growth medium supplemented by 500 µg/ml of Cycloheximide for 3 hr. This experiment was repeated on two occasions (independently cultured and plated imaging dishes)

Ver155008 incubation: Cells were incubated in normal growth medium supplemented by concentrations up to 50 µM in DMSO alongside DMSO control. Incubation was done for 3 hr and 9 hr.

Amino acid substitution: Cells were incubated for 3 hr in normal growth medium containing 5 mM Azetidine-2-carboxylic acid (Sigma Aldrich, USA). This stress was measured on its own on two separate occasions separate occasions (independently cultured and plated imaging dishes), and also as control dish for combination of amino acid substitution and proteasome inhibition on two separate occasions (independently cultured and plated imaging dishes).

RUVBL depletion: For siRNA transfection, we used Lipofectamine RNAiMAX (Invitrogen) and followed the manufacturer's reverse-transfection protocol. For a well on a 24-well plate, we mixed 0.4 µl of the reagent with 2 µl of 10 µM siRNA in 100 µl of OptiMEM and added the mixture to 400 µl of a cell suspension in the well. 24–28 hr later the transfection was stopped and the cells were plated for an experiment conducted the next day. We used the following siRNAs purchased from Dharmacon: siGENOME Non-Targeting siRNA #5 and siGENOME RUVBL1 siRNA (D-008977–04) this stress was measured on two separate occasions (independently cultured and plated imaging dishes).

## Cellular fixation

Fixation was carried out by 4% Paraformaldehyde (Electron Microscopy Sciences, USA) for 15 min at room temperature. In tests to investigate possible fixation artefacts we varied fixation Paraformaldehyde concentrations and temperatures including a variety of fixation timescales from fixation in. 05% paraformaldehyde at 4C for 16 hr to the conventional fixation for 15 min. Cells were then mostly imaged immediately after fixation but were never stored longer than 5 days before imaging. The storage was at 4C under Phosphate buffered saline in dark conditions. There was no noticeable difference between cells imaged immediately after fixation and a few days after. Different fixation rates showed no difference in the resulting cluster size distributions:

## Imaging
### Super-resolution imaging

For imaging, Cells were plated on 25 mm round glass coverslips (CS-25R) from Warner Instruments (Hamden, CT) for 12–24 hr in the specified growth conditions. Cells were either fixed and imaged or imaged live after reaching 50–75% confluence. All imaging was carried out in Leibowitz's L-15 medium. To conduct PALM Super resolution imaging, we used an optical system built using a Nikon Eclipse Ti microscope with a 100 × oil immersion objective (NA 1.40). Pre-converted Dendra2 was excited by a 488 nm laser line. Photo-activation of Dendra2 was carried out with a 405 nm laser line and in the post-converted state Dendra2 was excited by a 561 nm laser line. These laser lines were, expanded, re-collimated and focussed on the back focal plane of the Microscope in an external optical path using an achromatic beam expander (AC254-040-A and AC508-300-A, from THORLABS, Newton, NJ) and an achromatic converging lens (#45–354, from Edmund Optics, Barrington, NJ). Image data was collected using an Andor iXon Ultra 897 EMCCD camera. The laser power densities used for post-converted Dendra2 were 0.5 W/cm2 (405 nm) and 3.2 kW/cm2 (561 nm) on the image plane.

For live cell imaging the L15 media was supplemented with 10% Fetal Bovine Serum (Thermo Fisher). For both live and fixed cell imaging, the cells were maintained at 37°C in a temperature controlled platform (InVivo Scientific) on the microscope stage during image acquisition. Z-position of the microscope stage was maintained during acquisition using the Perfect Focus System (PFS) on the Nikon Ti Eclipse.

For fixed cell super-resolution imaging, movies with 10,000 frames, each averaged over 50 ms of exposure time, were acquired with both the excitation and photo-converting lasers on continuously.

## Lattice light sheet imaging

For Lattice Light Sheet Microscopy (*Chen et al., 2014*), cells were plated on gelatine-coated cover-slips 24–48 hr before imaging and grown as described above. Imaging took place in L15 medium supplemented with 10% FBS. A lattice light sheet consisting of 61 Bessel beams was generated with an annulus of inner/outer numerical aperture NA 0.44/0.55. Illumination power was 1.3 mW measured before the illumination objective. Volumetric image data with a temporal resolution of 15 s/volume (300 frames) was acquired by stepping cells through the light sheet in intervals of 0.3 μm with 50 ms exposure time using a sCMOS camera (Orca Flash v4.2, Hamamatsu). Images were processed (deskewed) using a modified version of MATLAB (The Mathworks) code supplied by *Chen et al. (2014)*. Analysis was performed on maximum intensity projections of images stacks.

## Super-resolution reconstruction

To identify single molecules in raw images of photo-converted Dendra2 fluorescence, the intensity signals were analysed using an adapted version of the multiple-target tracking algorithm (MTT) (*Sergé et al., 2008*) then we used our open software qSR for visualization, super-resolution reconstruction and DBSCAN. Briefly, for each frame, the point-spread function (PSF) of spatially separated individual fluorophores was detected and fitted to a two-dimensional Gaussian distribution. The centre of the fit yielded the position of single molecules with nanometre accuracy. Super-resolution reconstruction images were generated by superimposing a 2D Gaussian curve with the same intensity value central position and standard deviation as found by the fitting procedure. Finally, the positions of single molecules were fed into the DBSCAN (*Ester et al., 1996*) implementation custom written to extract meaningful distribution functions from the resulting data. Representative super resolved reconstructions along with zoom-ins showing DBSCAN efficacy in all measured conditions are shown in *Figure 1—figure supplement 1*.

## Quantification and analysis

### DBSCAN image analysis

DBSCAN, density-based scanning is a powerful computational technique for identifying correlations in a variety of data (*Ester et al., 1996*). Using two user chosen parameters, *m* and *r*, the algorithm combs through a data set of spatial coordinates – corresponding here to the super resolved localizations from Dendra2 – classifying the points as belonging to clusters if there are at least m points from that cluster within a radius r of the point. Our implementation was included in *Andrews et al. (2018)* and *Andrews et al. (2017)*.

Parameter choice will depend on the total density of localizations and the relative strength of local density fluctuations constituting clusters, both of these are influenced by imaging conditions. Therefore, parameters must be chosen by careful comparison to an unclustered control dataset acquired keeping total density of localizations as close to constant as possible. Across all data sets we consider the first 10,000 frames (50 ms integration time) for each cell in all experimental treatments and with constant imaging conditions.

We chose *r* as 40 nm and *m* as 10. Our choice of parameters was based on running the algorithm on cells transfected with plain Dendra2 as our unclustered control. While super-resolution maps with ~50000 localizations in ~$(25\mu m)^2$ area for Dendra2 cells gave between 50 and 150 clusters/cell, with the same localization density in Synphilin-Dendra2 cells we found of the order of 1000 clusters per cell. Changing r and m by a factor of 2 did not significantly affect the number of clusters found. Thus our parameter choice (and factors of two on either side of our parameter choice) was effectively eliminating noise and not missing clusters. Lastly, we visually inspected the localization maps and DBSCAN cluster allocations and never encountered a problem. Representative super resolved reconstructions along with zoom-ins showing DBSCAN efficacy in all measured conditions are shown in *Figure 1—figure supplement 1*.

## Analysis of cluster size distributions (Super-resolution)

Computation of the cluster size distribution functions was carried out in the following steps:

DBSCAN was run on 10,000 frames for each cell in a given experimental condition.

For each cluster identified by DBSCAN, the number of localizations making up the cluster and their spatial spread as estimated by drawing a convex hull around the points (Radius $R$) were tabulated. Data from all cells in similar conditions were collated. Since our uncertainty in super-resolved molecular positions is ~20 nm we discarded all clusters (collections of points with at least m = 10 neighbours) with diameter spanning less than 50 nm.

Next for each cluster we calculated the quantity $n = (R \ in \ nm/1nm)^3$. In each experimental condition, we have ~ 10000 clusters collated from ~ 10 cells. In data from untreated cells, > 90% of these clusters had $n$ values less than $n = 2 \times 10^6$. In highly clustered experimental data sets such as 180 minutes post proteasome inhibition, the proportion of large clusters increased. However, the majority of clusters were still less than $n = 2 \times 10^6$. The value of $n$ corresponding to a critical size was always less than $n = 2 \times 10^6$ depending on the experimental condition. The cluster size distribution functions in the main text and this supplement were fit to the theoretical form solely in the sub-critical range where the theory is valid (which is also where the majority of our cluster data lay).

The cluster size distribution functions in each experimental condition were computed from normalized histograms of the collated cluster $n$ values from that experimental condition with a constant bin size across all data sets ($\Delta n = 3 \times 10^4$). The main concern is to have enough data to be able to choose sufficiently small bins to effectively sample the fastest variations in the underlying distribution function without hitting a noise floor. This is the Nyquist criterion for binning. With more than 5000 clusters ranging for each condition from $n = 1.5 \times 10^4$ to $n = 2 \times 10^6$ (the sub-critical regime) we had sufficient data that even bins of size $\Delta n = 5000$– dividing the sub-critical regime into ~400 bins – resulted in histograms that were noiseless enough to fit and sampling the distribution very accurately. Data reported is with $\Delta n = 3 \times 10^4$ the results of fitting the distribution function are insensitive to changing the bin size down to $\Delta n = 5000$ (<2% change in estimated $R_c$).

For ease of presentation, we note that the probability distribution function is $Prob(n) = Ae^{-G(n)}$ where A enforces normalization. However A is determined by the fit parameters a and b since $\int_0^{n_c} Prob(n) = 1$ implies $= 1/\int_0^{n_c} e^{-G(n)}$. Then $-Log(Prob(n) = G(n) - Log(A)$, where Log refers to the natural log (base 'e'). Therefore, in order to read the nucleation barrier directly from the –Log(P(n) curve, we must add an offset $Log(A)$. Equivalently this amounts to a self-consistent normalization of our experimentally measured distribution function such that A is 1. This procedure just contributes an offset to the –Log(P(n)) curves without affecting critical radius and permits ease of presentation as it allows direct reading of the barrier height from the – Log(P(n)) graphs.

Fitting of the experimentally measured cluster size distribution functions to the theoretical functional form was carried out using the *Mathematica* implementation of least squares linear regression routine included in the *LinearModelFit* command. Errors in values of $R_c$ reported standard error of the mean from the best fit (computed, for instance using the *MeanConfidenceBand* object property of the *LinearModelFit* package in *Mathematica*).

The range of $n$ values for fitting data is important to determine. The theoretical form is only expected to hold below the critical size and diverges above the critical size. This sets an upper bound on range of data to use. However, the more data points you include the tighter the error bars on the fit parameters. The fitting range was determined self consistently by fitting data up to that value of $n$ which was 80% of the critical radius predicted by the fit. In practice, this amounted to fitting up to $n \sim 8 \times 10^5$ for the case of 180 min of proteasome inhibition (when the predicted critical size was $n \sim 1 \times 10^6$) and fitting up to $n \sim 1.5 \times 10^6$ for the case of untreated cells where the critical size corresponded to $n \sim 2 \times 10^6$. The larger error bars in the 3 hour proteasome inhibited case than in the untreated cell reflect both the greater spread in the data (fewer total clusters) and smaller available fitting range. The residuals in Fig. 1 depend slightly on the data range used to fit and subtract the $n^{2/3}$ term and consequently only qualitative conclusions (such as the sign of slope) should be drawn from relative comparisons between data sets fit identically. Our procedure was to always fit the range $n < 2 \times 10^5$ in all data sets and using bin size 5000 in this range.

The distribution of sub-critical cluster sizes P(n) is a Boltzmann distribution only in terms of the extensive variable $n_{tot}$, the total number of polypetides in the cluster. Our analysis has been in terms of the defined parameter $n = \left(\frac{R}{1 \ nm}\right)^3$. As defined, n should be proportional to the total number of molecules in the cluster as we image only a fraction of fluorescently detected molecules which

constitute only one species in our aggregates (*Figure 1—figure supplement 3*). However, the critical radius and nucleation barrier may be computed immune to a multiplicative constant in the definition of $n = \left(\frac{R}{1\,nm}\right)^3$. This can be seen by asking what would happen were we to misrepresent the cluster size $n_{tot}$ by an arbitrary multiplicative factor k. Then $n_{tot}$, the real cluster size would be replaced by $n' = kn_{tot}$. The free energy measured in terms of $n'$ would be $\Delta G(n') = -b'n' + a'n'^{2/3}$. The relation between primed parameters and the unprimed – true – parameters would be $b = b'k$ and $a = a'k^{2/3}$. Now the true value of the barrier height is $\Delta G(n_{tot_c}) = -bn_{tot_c} + a\,n_{tot_c}^{2/3}$ with $n_{tot}c = \left(\frac{2a}{3b}\right)^3$ the critical size. If we used the scaled variables b' and a' we would get

$$\Delta G\left(n_c'\right) = -b'n_c' + a'n_c'^{2/3} = -b'\left(\frac{2a'}{3b'}\right)^3 + a'\left(\left(\frac{2a'}{3b'}\right)^3\right)^{2/3}$$

On substitution, the factor k cancels out to yield $\Delta G(n_c') = -bn_c + a\,n_c^{\frac{2}{3}} = \Delta G(n_c)$. That is, even if we had cluster size parameter $n$ incorrect by a multiplicative factor, we can measure the barrier height (in terms of KT assuming ambient temperature as the relevant factor for thermalization). Similarly any multiplicative error made in converting R to n' can be shown to cancel when converting $n_c'$ back to Rc. For estimation of critical radius, we used the turning point of the fit function $\Delta G(n)$ to estimate critical size and then converted from $n_c$ to $R_c$ This procedure makes the determination of $R_c$ independent of any multiplicative factor in the definition of n as this same factor appears in $n_c$ and is cancelled out on going back from $n_c$ to $R_c$.

## Analysis of live cell (light sheet) data

Light sheet data was analyzed to extract both distribution of cluster intensity and time evolution.

- To calculate the intensity of the clusters in living cells, a three-dimension segmentation was performed using standard Mathematica functions (using the *MorphologicalComponents* and *ComponentsMeasurements* commands). For the instantaneous estimation of clusters sizes, only the first time point of each time series was analysed as this corresponds to the least photobleaching at this stage. To aid the segmentation, a 20-pixel background subtraction was performed on each image plane, however the actual intensity was calculated from the original unprocessed 3D stack using the total intensity within the segmented domain coordinates. Approximately 100–200 clusters were found in each cell.
- A control experiment was conducted by bleaching cells for 20 min when only single molecules (with single steps blinking and photo-bleaching) were visible – a 20 pixel background subtraction was used and the intensity of ~100 single molecules were measured. The resulting average single molecule intensity (24 counts) was used to convert the cluster intensities into estimated number of fluorescent molecules.
- The calculated cluster size converted into estimated number of fluorescent molecules was binned in bins of 10 (binsize = 20 did not change the shape of the curve but at binsize = 5 there was noticeably more noise) and plotted as in *Figure 4* in the main text and fit to the same functional form as for the fixed cell super-resolution data.
  To obtain the time trace of clusters sizes two challenges were considered: photo bleaching by the light sheet illumination, and large scale motion of clusters during the measurement interval.
- While the intensities in the first frame – used for the cluster size distribution of *Figure 4*- are unaffected by bleaching, the intensity values in subsequent frames – used in *Figure 5* – are affected. In order to correct for photo bleaching the average intensity of an imaging plane was measured as a function of time, and fit to an exponential. The exponential fit was used to correct the intensities at each time point. The data was acquired with very fine Z- steps (300 nm step size); larger step sizes caused the bleaching rate to be bi-exponential, putatively due to leaving Z-sections differentially bleached. We can see that our bleach correction is largely effective by noting that that different clusters in the same region of the same cell had different kinetics, some rising and some falling in intensity over the period of imaging.
- The motion of clusters during the measurement interval and in between time steps (15 s per cell stack) necessitated that we go through the bleach corrected light sheet movies, following individual clusters manually and using Mathematica (using the *MorphologicalComponents* and *ComponentsMeasurements* commands) based segmentation to identify coordinates for intensity calculations in ImageJ. The results from 30 such clusters are plotted in *Figure 5A,B* of the main text. This procedure resulted in studying clusters that were trackable for the whole movie.

## Acknowledgements

We thank Ammon Possey (WUSTL), Assaf Amitai (MIT), Ben Sabari (MIT), Gene Wei Li (MIT), Geraldine Seydoux (Johns Hopkins), Jeff Gore (MIT), Jeong-Mo Choi (WUSTL), Kabir Ramola (Brandeis), Kandice Tanner (NCI/NIH), Kiersten Ruff (WUSTL), Miccah Hecht (MIT), Rick Young (MIT), Rohit Pappu (WUSTL), Sina Wittman (MPI-CBG, Dresden), Tony Hyman (MPI-CBG, Dresden), Vivien Siegel (MIT) and members of the Cissé lab at MIT for helpful comments and discussions. Research reported in this publication was supported by the National Cancer Institute and the National Institutes of Health through the NIH Director's New Innovator Award Number DP2CA195769 to IIC. We acknowledge HHMI for the license to build Lattice Light Sheet. The content is solely the responsibility of the authors and does not necessarily represent the official views of the National Institutes of Health. This work was also supported by funds from the MIT Department of Physics.

## Additional information

### Funding

| Funder | Grant reference number | Author |
| --- | --- | --- |
| National Institutes of Health | DP2CA195769 | Ibrahim I Cisse |

The funders had no role in study design, data collection and interpretation, or the decision to submit the work for publication.

### Author contributions

Arjun Narayanan, Conceptualization, Software, Formal analysis, Investigation, Methodology, Writing—original draft, Writing—review and editing; Anatoli Meriin, Resources, Investigation, Methodology, Writing—review and editing; J Owen Andrews, Software, Investigation, Writing—review and editing; Jan-Hendrik Spille, Conceptualization, Software, Investigation, Methodology, Writing—review and editing; Michael Y Sherman, Resources, Investigation, Methodology, Project administration, Writing—review and editing; Ibrahim I Cisse, Conceptualization, Formal analysis, Supervision, Funding acquisition, Methodology, Writing—original draft, Project administration, Writing—review and editing

### Author ORCIDs

Arjun Narayanan http://orcid.org/0000-0002-2269-3253
Anatoli Meriin http://orcid.org/0000-0003-0087-814X
Ibrahim I Cisse http://orcid.org/0000-0002-8764-1809

### Decision letter and Author response

Decision letter https://doi.org/10.7554/eLife.39695.018
Author response https://doi.org/10.7554/eLife.39695.019

## Additional files

### Supplementary files

• Transparent reporting form
DOI: https://doi.org/10.7554/eLife.39695.016

### Data availability

All data generated or analyzed during this study are included in the manuscript and supporting files. Source data files have been provided for Figures 1, 2, 3 and 4.

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
