## [Decision Letter]

[**Editorial note:** This article has been through an editorial process in which the authors decide how to respond to the issues raised during peer review. The Reviewing Editor's assessment is that all the issues have been addressed.]

Thank you for submitting your article "A first order phase transition mechanism underlies protein aggregation in mammalian cells" for consideration by *eLife*. Your article has been reviewed by three peer reviewers, one of whom is a member of our Board of Reviewing Editors, and the evaluation has been overseen by Anna Akhmanova as the Senior Editor. The following individual involved in review of your submission has agreed to reveal his identity: Rohit Pappu (Reviewer #2). The other two reviewers remain anonymous.

The Reviewing Editor has highlighted the concerns that require revision and/or responses, and we have included the separate reviews below for your consideration. If you have any questions, please do not hesitate to contact us.

Using highly quantitative fluorescence imaging approaches, the authors characterize size distributions of clusters of misfolded proteins in unstressed mammalian cells. They show that the cluster size distribution is consistent with one that is expected for a super-saturated system, with clusters that continue to grow after they have formed. Further, the authors show that the protein RuvBL is at least partially responsible for a selective clearing of the large aggregates from the cells, while maintaining a super-saturated state (with the same size distribution of smaller aggregates). The reviewers express their excitement about the high quality of the work and how it is exemplary in applying physical principles to complex biological problems.

A key concern raised by the reviewers revolves around the clarity of the explanation of the physical models to a biological audience. Further, one of the reviewers raises a number of issues around potential artefacts of the drug treatments. Finally, one of the reviewers points out that the addition of a fluorescent protein to the synphilin might introduce changes in cluster-size distribution.

Separate reviews (please respond to each point):

*Reviewer #1:*

Using highly quantitative fluorescence imaging approaches, the authors show that clusters of misfolded proteins exist in unstressed mammalian cells, with a size distribution suggestive of misfolded proteins being present in a super-saturated phase: if a cluster is formed, it will keep growing. Question then is why unstressed cells remain clear of large aggregates. The authors show that the protein RuvBL is at least partially responsible for a selective clearing of the large aggregates from the cells, while maintaining a super-saturated state (with the same size distribution of smaller aggregates). They show that pharmaceutical methods of changing global levels of misfolded proteins result in changes in size distributions that affect all sizes, indicative of the system globally changing its super-saturation state.

There's a couple of methodological questions that should be addressed in a revised manuscript:

Key issue is that the authors need to show that expression of the FP-Synphilin marker does not change the size distribution for small aggregates. They rely on previously published work that shows that to be the case for large aggregates. Since the novelty of this work is primarily based on the ability to see the smaller aggregates, they'll need to put forward data or a strong argument for that to be true for all sizes.

Also, the authors relate fixed-cell data with live-cell information by equating volume distributions from fixed-cell measurements with fluorescence intensity distributions from live cells. That only works if the fluorescence scales with volume for the misfolded aggregates, something that cannot be assumed given the complex nature of binding of the Synphilin marker to the clusters.

*Reviewer #2:*

Amyloid formation, a long standing problem in molecular and cell biology with direct implications for neurodegeneration and molecular memory, and the problem of intracellular phase transitions are thought to be first-order phase transitions. Such transitions are governed by the presence of a saturation concentration and depending on the degree of supersaturation, the dynamics of phase separation – be it the formation of liquid-like condensates or fibrillar solids – will determine the interplay between nucleation vs. spinodal decomposition. These concepts are very well established in the physics literature. Are they germane to phase transitions in cells and what is the simplest theory that explains phase behavior in fixed / living cells? Narayanan et al. answer these questions using novel methods based on super resolution imaging of aggregation prone proteins in fixed / living cells. The striking result is that the tenets of homogeneous nucleation appear to apply rather shockingly well, albeit with a correction for cellular effects such as the clearance of higher-order, supercritical clusters by an active AAA+ ATP dependent enzyme. This work is striking and it represents the very best of science at the interface between physics and highly relevant biology. Taken together with recent work from the Brangwynne lab that is based on the use of optogenetic tools and the Halfmann lab based on so-called DAmFRET methods, the work of Narayanan et al. paves the way for serious investigations of the dynamics of phase transitions in living cells. The work is remarkable. The manuscript is very well written and easy to follow. And the findings are striking. *eLife* is fortunate to have received this manuscript and I urge publication pending a few important revisions that are detailed below.

Conceptual issues

1) The box that describes free energy expressions needs revision. The free energy of nucleation is written rather simply as ∆G = a*n^(2/3) {plus minus} bn. This is too coarse-grained and gives the impression of a purely phenomenological almost empirical treatment. In the mean-field description, the free energy of nucleation may be written precisely as ∆G = 4*π*γ*(R^2) – (4π/3)*rho_n*∆µ*(R^3). Here, R is the cluster radius, γ is the interfacial tension, ∆µ is the supersaturation (positive above the saturation concentration and negative below the saturation concentration), and rho_n is the number density of nucleation sites. Note that the sign of ∆µ sets the sign of the second term. Additionally, the magnitude of ∆µ sets the driving force for phase separation and hence the magnitude of the free energy barrier. I bring this up because the narrative does not clearly connect the pre-factors to the supersaturation and gives the impression that there is some ad hoc switch to be turned on or off. Importantly, the parameters can be further parsed by controlling the protein expression, which controls ∆µ. The authors need to be more explicit in the way they connect to the theory, i.e., start with the expressions in terms of R and then work back to the more compact form so the reader can see where the supersaturation fits in. Also, by not using the full expression, an opportunity is lost to probe how / why the homogeneous nucleation theory applies as well as it does.

2) The explanation of supersaturation needs work. As written, the verbiage in terms of liquid-vapor transitions, will be very confusing to the readership of *eLife*. It is best to use the parlance of phase separation rather than condensation transitions. The following references might be useful for the authors in terms of connecting to a more biological audience. Please see: https://www.sciencedirect.com/science/article/pii/S1097276518305033?_rdoc=1&_fmt=high&_origin=gateway&_docanchor=&md5=b8429449ccfc9c30159a5f9aeaa92ffb, http://www.pnas.org/content/110/50/20075.full.

3) The mathematics of the Szilard model could be made more explicit in the main text. There are concepts pertaining to the thermodynamics of preferential interactions and to the issue of competing timescales that should be discussed explicitly with regard to the Szilard model.

4) The Knowles group has argued, rather convincingly, that secondary nucleation is THE explanation for kinetics of amyloid formation (phase separation) in vitro. A detailed discussion of the connections to this work and the seemingly divergent views emerging from in cell studies would be helpful.

Scholarship issues

1) There is quite a bit of precedent, albeit from in vitro studies, for the types of observations made in the current study. There is the work of Garai et al. https://aip.scitation.org/doi/10.1063/1.2822322 showing the presence of sub- and supersaturated clusters for Abeta that is directly relevant and should be discussed. There is work from the Pappu lab predicting why there should be sub- and supersaturated clusters (http://dx.doi.org/10.1016/j.abb.2007.08.033; http://dx.doi.org/10.1016/j.jmb.2008.09.026), demonstrating that these exist for Httex1 in vitro (http://www.pnas.org/content/110/50/20075.full), and showing that there are distinct phase boundaries for Httex1 (http://www.jbc.org/content/293/10/3734). The latter work is particularly relevant because the S-phase species are likely to be first cousins of the supersaturated, sub-critical species documented by Narayanan et al. This is particularly relevant because Sathasivam et al. (see Figure 5 in http://dx.doi.org/10.1093/hmg/ddp467) show that these S-phase species are clearly prevalent in R6/2 mice brains expressing mutant Httex1. Finally, the recent work of Garai et al. shows that transthyretin can suppress the formation of fibrillar aggregates of Abeta by targeting and re-routing large, subcritical albeit supersaturated clusters. In the case of TTR, this is achieved by rerouting the molecules to amorphous phases whereas the Rblv5 molecules catalyze the clearance. Inasmuch as these are unifying themes, it would be useful to discuss these references and connect to them if at all possible.

There are quite a few pronouncements throughout that go without citation and / or lack citations to the primary literature. The Slezov book is helpful, but citations to the original papers of Szilard and Ostwald are essential.

In conclusion, this is an exciting paper, powered by innovations in super resolution imaging methods. With suitable revisions, this could make for an exceedingly exciting contribution to *eLife* – one that will generate rather a lot of discussion and help prompt a significant leap forward. The result showing that even under normal expression levels the system is supersaturated w.r.t. to the marker proteins is impressive. Prajwal Ciryam et al. (working with Rick Morimoto, Chris Dobson, and Michele Vendruscolo) have identified the fraction of the proteome that should be supersaturated under normal expression levels. This work seems particularly relevant in the current context.

Minor Comments:

On a semantic note, the theory that authors are using pertains to homogeneous nucleation not simple nucleation or just nucleation. The correct terminology should be used and the distinctions between homogeneous vs. heterogeneous nucleation and non-nucleated processes should be articulated clearly.

The authors seem to be using a rather primitive equation editor and some of the equations are hard to follow. This could be cleaned up.

A purely persnickety point: Data are plural and datum is singular. Please fix this issue, which shows up throughout.

There also are rather a lot of typos that should be cleaned up.

Additional data files and statistical comments:

It would be very helpful to have access to the raw data for the clusters thus enabling independent analysis and comparative assessments for those who are interested.

*Reviewer #3:*

This manuscript describes a mechanism and new model for the aggregation process of synphilin and α-synuclein in cell culture. In essence the oligomerization behaves like a phase transition of monomers into protein condensates but whereby larger clusters are selectively removed from the system in a non-equilibrium flux as new monomers are produced (i.e., the Szilard model). The authors examined the impact of drug treatments that affect protein production and aggregation, as well as putative chaperone RuvBL to test the models.

Comments and points

1) The authors have clearly spent effort to explain the details of the physics underpinning the models and I think the application of physics is a real strength of the work. The great challenge for me as a non-physicist is that it is not so easy to understand the underlying basis of the theory – so I cannot comment on whether the models are appropriate or consistent with the actual data and will leave that to other reviewers with the relevant background.

2) It seems a little counterintuitive to me that expression level of synphilin and synuclein does not effect the clustering patterns (Figure 1—figure supplement 2) because for many proteins that form aggregates in cells (eg polyglutamine) expression level is a major parameter affecting aggregation. Perhaps what is meant here is that the expression of these proteins are used as tracers for pre-existing aggregates present in the cell. If so, that needs to be rephrased more clearly. If not, does that mean the models used here are not representative of other systems?

3) The treatment with rapamycin is not as straightforward as the authors indicate. Rapamycin inhibits mTOR which will affect a wide range of cellular processes including autophagy, which will affect the clearance mechanisms of proteins (Li, J., Sang G. Kim, and J. Blenis, Rapamycin: One Drug, Many Effects. Cell Metabolism, 2014. 19(3): p. 373-379.) A more direct inhibitor of translation would be cycloheximide, which directly inhibits the ribosome. This really should be investigated if translation is to be properly examined.

4) Also, do the drug treatments change the levels of synphilin? (which I would anticipate they do) and if so, how would this affect the modelling? Also, there is a reasonable chance that exerting stress on these cells (certainly with MG132) can lead to the spontaneous formation of stress granules or other protein condensates (and indeed other stress responses that alter the broader activity of the protein quality control systems). How does the model account for cell-regulated formation of such structures that might coalesce with synphilin, or upregulation of stress responses?

5) The results with RuvBL are intriguing. Nonethless RuvBL does seem a left-field choice of "chaperone" since much of the literature points to it being involved as a DNA helicase. The mechanisms might be more compelling if Hsp70-mediated mechanisms were also examined given that these are classic systems for overseeing protein folding, triage mechanisms and in dissolution of protein aggregates. (For example, Gao et al, 2015). It would be rather straightforward to test these mechanisms alongside. For example, specific Hsp70 family inhibitors are also available and would be very interesting to test here (eg VER-155008).

Minor Comments:

1) In the Introduction where the theory of first order phase transitions is described, I suggest it be rephrased slightly for the sake of the biology readers so that it is explicitly stated that this is a classic model being extrapolation to the solution protein context (whereby proteins are not obviously not in a gas phase).

2)The sentence "Because cells under normal growth conditions do not show large growing clusters, the naïve hypothesis is that the cell is normally in a sub-saturated state" is unclear from a biological perspective because many or most proteins normally exist and function as oligomers or as part of large complexes.

3) The sentence "Our observation is both surprising and intriguing as the Szilard model was not previously thought to exist in a natural system (Slezov, 2009),". While I understand the point the authors are making, this statement needs to be more properly discussed in context of well-understood biological processes that effectively behave like the Szilard model. In other words, there are very well-established concepts and mechanisms that govern protein homeostasis in terms of protein production and degradation (eg degradation of protein aggregates by autophagy; a review on this topic from Bukau's lab: Tyedmers, J., A. Mogk, and B. Bukau, Cellular strategies for controlling protein aggregation. Nature Reviews Molecular Cell Biology, 2010. 11(11): p. 777-788.).

4) Referring to the sentences: "How Synphilin1 is recruited to aggregates is not fully understood. However this protein is a commonly used marker for well-studied misfolded protein aggregates such as aggresomes and Lewy bodies (Tanaka et al., 2004; Wakabayashi et al., 2000) and the ectopic expression, and the expression levels have no detectable effect on the formation of the aggresome (Zaarur et al., 2008)." It is not clear what is meant by this phrase since normally aggresomes are only present in cell culture models when proteins are expressed abundantly and certainly in an ectopic setting. Also, the aggresome as a model is fraught with difficulties – please see this review for a detailed discussion on why this is the case (Radwan, M., R.J. Wood, X. Sui, and D.M. Hatters, When proteostasis goes bad: Protein aggregation in the cell. IUBMB Life, 2017. 69(2): p. 49-54.).

---

## [Author Response]

A key concern raised by the reviewers revolves around the clarity of the explanation of the physical models to a biological audience. Further, one of the reviewers raises a number of issues around potential artefacts of the drug treatments. Finally, one of the reviewers points out that the addition of a fluorescent protein to the synphilin might introduce changes in cluster-size distribution.Separate reviews (please respond to each point):

Reviewer #1:

Using highly quantitative fluorescence imaging approaches, the authors show that clusters of misfolded proteins exist in unstressed mammalian cells, with a size distribution suggestive of misfolded proteins being present in a super-saturated phase: if a cluster is formed, it will keep growing. Question then is why unstressed cells remain clear of large aggregates. The authors show that the protein RuvBL is at least partially responsible for a selective clearing of the large aggregates from the cells, while maintaining a super-saturated state (with the same size distribution of smaller aggregates). They show that pharmaceutical methods of changing global levels of misfolded proteins result in changes in size distributions that affect all sizes, indicative of the system globally changing its super-saturation state.

We are grateful to the reviewer for their careful reading of our manuscript and comments, which highlighted to us ways in which we could improve the clarity of our presentation. We address the reviewer’s specific comments below.

There's a couple of methodological questions that should be addressed in a revised manuscript:Key issue is that the authors need to show that expression of the FP-Synphilin marker does not change the size distribution for small aggregates. They rely on previously published work that shows that to be the case for large aggregates. Since the novelty of this work is primarily based on the ability to see the smaller aggregates, they'll need to put forward data or a strong argument for that to be true for all sizes.

As the reviewer points out, it is necessary to show that the size distribution of small aggregates does not depend on the expression level of Synphilin marker protein. In the revised manuscript, this is presented in Figure 1—figure supplement 2, in this figure we show that low Synphilin expression cells and high Synphilin expression cells have indistinguishable size distributions. We attribute this to the fact that Synphilin in our system is acts as a tracer for the multi-component clusters. We refer to this in the main text for example in the sentence:

“We checked that neither the expression level of Synphilin1 tracer protein nor the identity of the tracer (alternative tracer α-synuclein) have any detectable effect on the size distribution of subdiffractive clusters (Figure 1—figure supplement 2). This suggests that Synphilin1 in our sub-diffractive clusters merely serves as a tracer and does not on its own affect cluster formation at the expression levels tested.”

Also, the authors relate fixed-cell data with live-cell information by equating volume distributions from fixed-cell measurements with fluorescence intensity distributions from live cells. That only works if the fluorescence scales with volume for the misfolded aggregates, something that cannot be assumed given the complex nature of binding of the Synphilin marker to the clusters.

It is in fact the case that fluorescence detections scales with volume (cluster radius cubed), and we apologize for not making this point clearer: This analysis is in Figure 1—figure supplement 3 of the revised manuscript. It is true that the distribution of synphilin marker inside the clusters could have been more complex. However, when we investigate the distribution of synphilin in the clusters we find that the number of detections scales with the volume of the cluster. In fact this was one the first indication for us that clusters of had a well-defined density (a tale-tell sign of condensates).

We have now modified the text to read:

“We find that the number of localization events in a cluster, scales with the cube of the measured cluster radius This suggest that, at the relevant cluster sizes, the fluorescent detection events of the Synphilin1 tracer protein may be spread throughout the cluster volume at uniform density (Figure 1—figure supplement 3).”

Reviewer #2:

Amyloid formation, a long standing problem in molecular and cell biology with direct implications for neurodegeneration and molecular memory, and the problem of intracellular phase transitions are thought to be first-order phase transitions. Such transitions are governed by the presence of a saturation concentration and depending on the degree of supersaturation, the dynamics of phase separation – be it the formation of liquid-like condensates or fibrillar solids – will determine the interplay between nucleation vs. spinodal decomposition. These concepts are very well established in the physics literature. Are they germane to phase transitions in cells and what is the simplest theory that explains phase behavior in fixed / living cells? Narayanan et al. answer these questions using novel methods based on super resolution imaging of aggregation prone proteins in fixed / living cells. The striking result is that the tenets of homogeneous nucleation appear to apply rather shockingly well, albeit with a correction for cellular effects such as the clearance of higher-order, supercritical clusters by an active AAA+ ATP dependent enzyme. This work is striking and it represents the very best of science at the interface between physics and highly relevant biology. Taken together with recent work from the Brangwynne lab that is based on the use of optogenetic tools and the Halfmann lab based on so-called DAmFRET methods, the work of Narayanan et al. paves the way for serious investigations of the dynamics of phase transitions in living cells. The work is remarkable. The manuscript is very well written and easy to follow. And the findings are striking. eLife is fortunate to have received this manuscript and I urge publication pending a few important revisions that are detailed below.

We thank the reviewer for these effusive comments and we are delighted by the positive reception. Their careful reading, accurate criticism and help suggestions have helped to improve clarity and presentation.

The review included multiple suggestions and comments, which we address below.

Conceptual issues1) The box that describes free energy expressions needs revision. The free energy of nucleation is written rather simply as ∆G = a*n^(2/3) {plus minus} bn. This is too coarse-grained and gives the impression of a purely phenomenological almost empirical treatment. In the mean-field description, the free energy of nucleation may be written precisely as ∆G = 4*π*γ*(R^2) – (4π/3)*rho_n*∆µ*(R^3). Here, R is the cluster radius, γ is the interfacial tension, ∆µ is the supersaturation (positive above the saturation concentration and negative below the saturation concentration), and rho_n is the number density of nucleation sites. Note that the sign of ∆µ sets the sign of the second term. Additionally, the magnitude of ∆µ sets the driving force for phase separation and hence the magnitude of the free energy barrier. I bring this up because the narrative does not clearly connect the pre-factors to the supersaturation and gives the impression that there is some ad hoc switch to be turned on or off. Importantly, the parameters can be further parsed by controlling the protein expression, which controls ∆µ. The authors need to be more explicit in the way they connect to the theory, i.e., start with the expressions in terms of R and then work back to the more compact form so the reader can see where the supersaturation fits in. Also, by not using the full expression, an opportunity is lost to probe how / why the homogeneous nucleation theory applies as well as it does.

The reviewer’s point is well taken, and the box is now substantially revised as a result. In an attempt to make the theory accessible to a biological audience, and keeping in mind other subtle considerations (e.g. the specific pre-factors the reviewer indicates are dependent on geometric assumptions, and R not being an extensive variable) we have revised the description in the box including also clear statements of how the free energy function is determined by the interactions and concentrations within the system.

2) The explanation of supersaturation needs work. As written, the verbiage in terms of liquid-vapor transitions, will be very confusing to the readership of eLife. It is best to use the parlance of phase separation rather than condensation transitions. The following references might be useful for the authors in terms of connecting to a more biological audience. Please see: https://www.sciencedirect.com/science/article/pii/S1097276518305033?_rdoc=1&_fmt=high&_origin=gateway&_docanchor=&md5=b8429449ccfc9c30159a5f9aeaa92ffb, http://www.pnas.org/content/110/50/20075.full.

Based on the reviewer’s suggestion, we have now explicitly indicated the analogy to phase separation, in addition to including more of the literature. In the revised manuscript, the related sentence (and several similar phrases throughout the manuscript) now reads:

“A first order phase transition describes the discontinuous changes needed for a system to go from a dispersed phase to a condensed phase (or vice versa). This may correspond to the concentration of a single component from its dispersed phase (for example condensation) or the demixing of some components from a multicomponent mixture (for example liquid-liquid phase separation).”

3) The mathematics of the Szilard model could be made more explicit in the main text. There are concepts pertaining to the thermodynamics of preferential interactions and to the issue of competing timescales that should be discussed explicitly with regard to the Szilard model.

In response to the reviewer’s comment here (and to their suggestion in #1) we have now introduce the Szilard model in the box, concurrent with the mathematics and discussion of super-saturation and its relation to the interactions and ambient concentrations.

*4) The Knowles group has argued, rather convincingly, that secondary nucleation is THE explanation for kinetics of amyloid formation (phase separation)* in vitro*. A detailed discussion of the connections to this work and the seemingly divergent views emerging from in cell studies would be helpful.*

This is better addressed concurrently with the reviewer’s next suggestion. The discussion now reads:

“There is mounting evidence in the literature suggesting that diseased state aggregates may proceed first through the type of nucleation process that we observe here. In a classic study (Serio et al., 2000), Lindquist and colleagues examined the in vitro mechanism by which a prion protein Sup35 forms self-seeded amyloid fibers. They found that variably sized spherical complexes (akin to condensates and apparent by high resolution imaging (STEM &AFM)), appear to be crucial intermediates in de novo amyloid nucleation (Serio et al., 2000). Similarly for Huntingtin N terminal fragments, in vitro studies (Crick et al., 2013; Posey et al., 2018) show that below the concentration required for growth of a structured fibrilar phase, there exists another saturation point distinguishing monomers and small oligomers of <10nm size from spherical complexes of ~25nm size (Posey et al., 2018); globular complexes have also been isolated from Huntingtin rat models (Sathasivam et al., 2010). Intermediate steps are also suggested by fluorescence correlation spectroscopy studies of nucleation of amyloids, where hints of a second nucleation barrier between small multimers and large structured aggregates were seen (Garai et al., 2008). In light of such precedence and the concentrations needed for fibril formation in vivo, it is plausible that, the micro-environment within in vivo condensates such as the ones found in our current study, could serve to provide the high concentrations needed for the secondary nucleation (Törnquist et al., 2018) and maturation of clusters with various properties.”

Scholarship issues

*1) There is quite a bit of precedent, albeit from* in vitro *studies, for the types of observations made in the current study. There is the work of Garai et al. https://aip.scitation.org/doi/10.1063/1.2822322 showing the presence of sub- and supersaturated clusters for Abeta that is directly relevant and should be discussed. There is work from the Pappu lab predicting why there should be sub- and supersaturated clusters (http://dx.doi.org/10.1016/j.abb.2007.08.033; http://dx.doi.org/10.1016/j.jmb.2008.09.026), demonstrating that these exist for Httex1* in vitro *(http://www.pnas.org/content/110/50/20075.full), and showing that there are distinct phase boundaries for Httex1 (http://www.jbc.org/content/293/10/3734). The latter work is particularly relevant because the S-phase species are likely to be first cousins of the supersaturated, sub-critical species documented by Narayanan et al. This is particularly relevant because Sathasivam et al. (see Figure 5 in http://dx.doi.org/10.1093/hmg/ddp467) show that these S-phase species are clearly prevalent in R6/2 mice brains expressing mutant Httex1. Finally, the recent work of Garai et al. shows that transthyretin can suppress the formation of fibrillar aggregates of Abeta by targeting and re-routing large, subcritical albeit supersaturated clusters. In the case of TTR, this is achieved by rerouting the molecules to amorphous phases whereas the Rblv5 molecules catalyze the clearance. Inasmuch as these are unifying themes, it would be useful to discuss these references and connect to them if at all possible.*

We have added the suggested literature to our previous discussion. This part now reads:

“There is mounting evidence in the literature suggesting that diseased state aggregates may proceed first through the type of nucleation process that we observe here. In a classic study (Serio et al., 2000), Lindquist and colleagues examined the in vitro mechanism by which a prion protein Sup35 forms self-seeded amyloid fibers. They found that variably sized spherical complexes (akin to condensates and apparent by high resolution imaging (STEM &AFM)), appear to be crucial intermediates in de novo amyloid nucleation (Serio et al., 2000). Similarly for Huntingtin N terminal fragments, in vitro studies (Crick et al., 2013; Posey et al., 2018) show that below the concentration required for growth of a structured fibrilar phase, there exists another saturation point distinguishing monomers and small oligomers of <10nm size from spherical complexes of ~25nm size (Posey et al., 2018); globular complexes have also been isolated from Huntingtin rat models (Sathasivam et al., 2010). Intermediate steps are also suggested by fluorescence correlation spectroscopy studies of nucleation of amyloids, where hints of a second nucleation barrier between small multimers and large structured aggregates were seen (Garai et al., 2008). In light of such precedence and the concentrations needed for fibril formation in vivo, it is plausible that, the micro-environment within in vivo condensates such as the ones found in our current study, could serve to provide the high concentrations needed for the secondary nucleation (Törnquist et al., 2018) and maturation of clusters with various properties.”

There are quite a few pronouncements throughout that go without citation and / or lack citations to the primary literature. The Slezov book is helpful, but citations to the original papers of Szilard and Ostwald are essential.

We apologize for this oversight and have now cited the original published literature. Several ideas (esp. originating from Leo Szilard) in nucleation are recorded by other authors, which makes it seem as though citations to their original papers are missing.

The original reference to Ostwald’s paper (Ostwald, 1897) was included. For nucleation in super-saturation we cited the earliest relevant literature (Farkas, 1927). We note that the original idea on the “steady state super-saturation” is generally “attributed to Szilard” by various authors, but to our knowledge Leo Szilard did not explicitly write it down as an authored publication. For instance, in (Farkas, 1927) the author explicitly cites “Die idee […] stammt von L. Szilard” attributing that the ideas they are proving about nucleation and size distribution originated from Leo Szilard, without Szilard as an author.

In conclusion, this is an exciting paper, powered by innovations in super resolution imaging methods. With suitable revisions, this could make for an exceedingly exciting contribution to eLife – one that will generate rather a lot of discussion and help prompt a significant leap forward. The result showing that even under normal expression levels the system is supersaturated w.r.t. to the marker proteins is impressive. Prajwal Ciryam et al. (working with Rick Morimoto, Chris Dobson, and Michele Vendruscolo) have identified the fraction of the proteome that should be supersaturated under normal expression levels. This work seems particularly relevant in the current context.

We thank the reviewer again for the laudatory comments and helpful suggestion.

Minor Comments:On a semantic note, the theory that authors are using pertains to homogeneous nucleation not simple nucleation or just nucleation. The correct terminology should be used and the distinctions between homogeneous vs. heterogeneous nucleation and non-nucleated processes should be articulated clearly.

We now introduce the theory as homogenous nucleation theory as suggested. In the Introduction section now reads:

“The details of classical nucleation theory are revealing about how cells maintain homeostasis. The so-called homogenous nucleation is a prototypical mechanism by which first order phase transitions proceed (Kalikmanov, 2013; Sear, 2007; Slezov, 2009)”

The authors seem to be using a rather primitive equation editor and some of the equations are hard to follow. This could be cleaned up.A purely persnickety point: Data are plural and datum is singular. Please fix this issue, which shows up throughout.There also are rather a lot of typos that should be cleaned up.

We apologize for these formatting issues; they should now have been fixed.

Additional data files and statistical comments:It would be very helpful to have access to the raw data for the clusters thus enabling independent analysis and comparative assessments for those who are interested.

All data files for all the figures in the text are made available online on *eLife* website as per the journal policy.

Reviewer #3:

This manuscript describes a mechanism and new model for the aggregation process of synphilin and α-synuclein in cell culture. In essence the oligomerization behaves like a phase transition of monomers into protein condensates but whereby larger clusters are selectively removed from the system in a non-equilibrium flux as new monomers are produced (i.e., the Szilard model). The authors examined the impact of drug treatments that affect protein production and aggregation, as well as putative chaperone RuvBL to test the models.

We thank the reviewer for their detailed reading of the manuscript. We are happy the reviewer as a non-physicist still appreciated the efforts made to explain detailed physics concepts underlying our findings. We are also grateful to the reviewer for their many suggestions of how to improve the clarity and for suggestions of additional biological controls.

Comments and points1) The authors have clearly spent effort to explain the details of the physics underpinning the models and I think the application of physics is a real strength of the work. The great challenge for me as a non-physicist is that it is not so easy to understand the underlying basis of the theory – so I cannot comment on whether the models are appropriate or consistent with the actual data and will leave that to other reviewers with the relevant background.

We thank the reviewer. By keeping in mind this comment, and utilizing suggestions from Reviewer 2 we have reworked physical description throughout the text, and hope to have made it even more accessible to a non-physicist readership.

2) It seems a little counterintuitive to me that expression level of synphilin and synuclein does not effect the clustering patterns (Figure 1—figure supplement 2) because for many proteins that form aggregates in cells (eg polyglutamine) expression level is a major parameter affecting aggregation. Perhaps what is meant here is that the expression of these proteins are used as tracers for pre-existing aggregates present in the cell. If so, that needs to be rephrased more clearly. If not, does that mean the models used here are not representative of other systems?

The reviewer is correct; we mean to indicate that synphilin acts as a tracer. We had previous used the language “marker” but have now changed them to “tracers” based on this reviewer’s comment. We further address this in response to reviewer’s comment 4. The text now reads:

“This suggests that Synphilin1 in our sub-diffractive clusters merely serves as a tracer and does not on its own affect cluster formation at the expression levels tested.”

3) The treatment with rapamycin is not as straightforward as the authors indicate. Rapamycin inhibits mTOR which will affect a wide range of cellular processes including autophagy, which will affect the clearance mechanisms of proteins (Li, J., Sang G. Kim, and J. Blenis, Rapamycin: One Drug, Many Effects. Cell Metabolism, 2014. 19(3): p. 373-379.) A more direct inhibitor of translation would be cycloheximide, which directly inhibits the ribosome. This really should be investigated if translation is to be properly examined.

We agree with the reviewer and have included in a new supplementary figure the results of our tests with Cycloheximide, which corroborates our original conclusions. We include the result of this test – which successfully showed a lowering of aggregation propensity compared to control cells – in Figure 2—figure supplement 2 and referred to it the main text.

4) Also, do the drug treatments change the levels of synphilin? (which I would anticipate they do) and if so, how would this affect the modelling? Also, there is a reasonable chance that exerting stress on these cells (certainly with MG132) can lead to the spontaneous formation of stress granules or other protein condensates (and indeed other stress responses that alter the broader activity of the protein quality control systems). How does the model account for cell-regulated formation of such structures that might coalesce with synphilin, or upregulation of stress responses?

We expect the drug treatment to affect all de novo protein synthesis; however the expression level of tracer Synphilin did not affect the measured clusters (Figure 1—figure supplement 2). We estimate that the labeled tracer molecule occupies at most one thousandth of the volume of the cluster, so we do not expect the tracer expression itself to affect our model, unless it significantly changed the total super-saturation of the cells.

It is possible that different perturbations or stress can have differing or even competing effect biologically; however, a “beauty” in the physical model is that the net effect of any perturbation should depend on the resulting ambient concentration of aggregating monomers. Thus if the net effect is such that the super-saturation (i.e. ambient concentration of monomers) is increased, then regardless of the cause this would result in a lower nucleation barrier (and smaller critical cluster radius); and that is for example the case for MG and AZC treatments. If the net effect of a perturbation is such the super-saturation is decreased, this results in higher nucleation barrier (and greater critical cluster radius); and that is for example what we see cyclohexamide and rapamycin. In the main text this conclusion now reads:

“Therefore, over a range of complex pharmacological perturbations –with simple intuition of how the perturbation would affect the saturation– the measured effects match the expectation from classical nucleation theory.”

5) The results with RuvBL are intriguing. Nonethless RuvBL does seem a left-field choice of "chaperone" since much of the literature points to it being involved as a DNA helicase. The mechanisms might be more compelling if Hsp70-mediated mechanisms were also examined given that these are classic systems for overseeing protein folding, triage mechanisms and in dissolution of protein aggregates. (For example, Gao et al., 2015). It would be rather straightforward to test these mechanisms alongside. For example, specific Hsp70 family inhibitors are also available and would be very interesting to test here (eg VER-155008).

We thank the reviewer for this suggestion, and have now examined Hsp70 mediated mechanism in this context. We tested whether inhibiting HSP70 with VER155008 could result in similar observations as with RuvBL knockdown (i.e. accumulation of super-critical clusters). However at the conditions we tried, Ver155008 treatment resulted did NOT prevent the putative clearance of super-critical clusters (but also did not change the distribution of sub-critical clusters). This suggests to us that Hsp70-mediated mechanism may be distinct from the RuvBL clearance. We have discussed this result in the main text and added the data in Figure 2—figure supplement 1.

Minor Comments:1) In the Introduction where the theory of first order phase transitions is described, I suggest it be rephrased slightly for the sake of the biology readers so that it is explicitly stated that this is a classic model being extrapolation to the solution protein context (whereby proteins are not obviously not in a gas phase).

We clarified the potential misunderstanding the reviewer raises by replacing everywhere the word “gaseous phase” by the word ‘disperse phase’ we also include explicit phrasing to describe how, as the reviewer very clearly points out we are applying ‘thermodynamic concepts to the solution protein context’. While doing this we also kept in mind the suggestions of Reviewers 1 and 2 and so included phrasing describing the two contexts that may be familiar to audiences (e.g. condensation and phase separation). For example the sentence now reads:

“A first order phase transition describes the discontinuous changes needed for a system to go from a dispersed phase to a condensed phase (or vice versa). This may correspond to the concentration of a single component from its dispersed phase (for example condensation) or the demixing of some components from a multicomponent mixture (for example liquid-liquid phase separation).”

2) The sentence "Because cells under normal growth conditions do not show large growing clusters, the naïve hypothesis is that the cell is normally in a sub-saturated state" is unclear from a biological perspective because many or most proteins normally exist and function as oligomers or as part of large complexes.

We apologize for the confusion. Here by ‘large growing clusters’ we mean “super-critical” clusters (we estimate N~1500 tracer molecules, which is orders of magnitude more than typical protein oligomers and protein complexes)

3) The sentence "Our observation is both surprising and intriguing as the Szilard model was not previously thought to exist in a natural system (Slezov, 2009),". While I understand the point the authors are making, this statement needs to be more properly discussed in context of well-understood biological processes that effectively behave like the Szilard model. In other words, there are very well-established concepts and mechanisms that govern protein homeostasis in terms of protein production and degradation (eg degradation of protein aggregates by autophagy; a review on this topic from Bukau's lab: Tyedmers, J., A. Mogk, and B. Bukau, Cellular strategies for controlling protein aggregation. Nature Reviews Molecular Cell Biology, 2010. 11(11): p. 777-788.).

We agree with the reviewer. We have now removed this comment.

4) Referring to the sentences: "How Synphilin1 is recruited to aggregates is not fully understood. However this protein is a commonly used marker for well-studied misfolded protein aggregates such as aggresomes and Lewy bodies (Tanaka et al., 2004; Wakabayashi et al., 2000) and the ectopic expression, and the expression levels have no detectable effect on the formation of the aggresome (Zaarur et al., 2008)." It is not clear what is meant by this phrase since normally aggresomes are only present in cell culture models when proteins are expressed abundantly and certainly in an ectopic setting. Also, the aggresome as a model is fraught with difficulties – please see this review for a detailed discussion on why this is the case (Radwan, M., R.J. Wood, X. Sui, and D.M. Hatters, When proteostasis goes bad: Protein aggregation in the cell. IUBMB Life, 2017. 69(2): p. 49-54.).

We have reworked this in an attempt to be clearer: The aggresome is not the subject of our study, and we take the reviewer’s point that the aggresome as a model system has difficulties (as listed in the reference above) which are unrelated to the sub-diffractive condensates studied here. We have edited the lines concerned to be clearer including removing reference to aggresome and highlighting the synphilin 1 traced aggregates as our subject of interest. This part now reads:

“How Synphilin1 is recruited to aggregates is not fully understood. However this protein is a commonly used tracer for well-studied misfolded protein aggregates such as Lewy bodies (Tanaka et al., 2004; Wakabayashi et al., 2000). Here we concentrate on sub-diffractive Synphilin1 traced aggregates whose size distribution we measure.”